

# Sensitivity of hydrological model to the temporal and spatial resolutions of rainfall input

Yingchun Huang[1], András Bárdossy[2], and Ke Zhang[1,3]

[1]College of Hydrology and Water Resources, Hohai University, Nanjing 210098, China
[2]Institute for Modelling Hydraulic and Environmental Engineering, University of Stuttgart, Stuttgart D-70596, Germany
[3]State Key Laboratory of Hydrology-Water Resources and Hydraulic Engineering, Hohai University, Nanjing 210098, China
**Correspondence:**
Ke Zhang (kzhang@hhu.edu.cn); Yingchun Huang(yingchunhuang@hhu.edu.cn)

**Abstract.** As the most important input for rainfall-runoff models, precipitation is usually observed at specific sites on a daily or sub-daily time scale and requires interpolation for further application. This study aims to explore that for a given objective function, whether a higher temporal and spatial resolution of precipitation could provide an improvement in model performance. Four different gridded hourly and daily precipitation datasets, with a spatial resolution of $1 \times 1$ km$^2$ for the Baden-Württemberg

state of Germany, were constructed using a combination of data from a dense network of daily rainfall stations and a less dense network of pluviometers with high temporal-resolution rainfall observations. Two different flavors of HBV models with different model structures, lumped and spatially distributed, were used to test the sensitivity of model performance on the spatial resolution of precipitation. For four selected mesoscale catchments located at the upstream region of Baden-Württemberg, these four precipitation datasets were used to simulate the daily discharges using both lumped and semi-distributed HBV mod-

els. Different possibilities of improving the accuracy of daily streamflow prediction were investigated. Three main results were obtained from this study: (1) a higher temporal resolution of precipitation improved the model performance if the observation density was high; (2) a combination of observed high temporal-resolution observations with disaggregated daily precipitation leads to a further improvement in the model performance; (3) for the present research, the increase of spatial resolution improved the performance of the model insubstantially or only marginally for most of the study catchments.

## 1   Introduction

Conceptual hydrological models have been developed to represent dynamic response of a particular catchment resulting from meteorological driving forces (Hundecha et al., 2008). Among meteorological variables, precipitation, which is traditionally measured using rain gauges, has a direct and crucial impact on the runoff response of a catchment (Obled et al., 1994; Ly et al., 2013). However, uncertainty in capturing the variability of precipitation by the rain gauges or wireless telemetering

constitutes a significant source of uncertainty for hydrological modeling (Berne et al., 2004). Previous studies have shown that hydrological models are sensitive to the observation network density and data quality (Singh, 1997; Kobold and Brilly, 2006; Bardossy and Das, 2008; Xu et al., 2013). Therefore, the precipitation input should be as accurate as possible to achieve better rainfall-runoff simulation and model parameter estimation (Cole and Moore, 2008; Ficchi et al., 2016).





Many research efforts have been carried out in the recent years for interpolating spatially distributed rainfall datasets (Goovaerts, 2000; Jeffrey et al., 2001; Hofierka et al., 2002; Haylock et al., 2008; Ly et al., 2013), as well as for the sub-daily disaggregation of daily rainfall (Parkes et al., 2013; Bardossy and Pegram, 2016). These approaches can potentially improve the quality and resolution of the precipitation data that are used as input for rainfall-runoff models, thereby reducing

the uncertainty of hydrological models. By design, most of the hydrological models are flexible and can be easily adjusted to different time steps of input datasets. Hydrological models are normally classified as lumped or distributed, depending on the degree of spatial discretization when describing the catchment (Ly et al., 2013). Bruneau et al. (1995) indicated the temporal and spatial resolutions used for the inputs of the hydrological model have an important influence on the model performance. Kobold and Brilly (2006) suggested that calibrating hydrological models with sub-daily time steps can significant improve

flood forecasting. Das et al. (2008) used different model structures to simulate daily runoff in the region of central Europe and showed that semi-distributed model structure could outperform lumped model structure.

The aim of this study is to gain knowledge on the dependency of hydrological model performance on the precipitation data. The effects of rainfall data quality on model performance were investigated. The sensitivity of model performance to different spatial and temporal resolutions of rainfall data was examined using two different model structures. The possibility of

improving model performance on daily scale was discussed. The manuscript is organized as follows: the introduction, followed by section 2, which describes the study area and the precipitation datasets used in this research. In section 3, the hydrological model and the calibration framework used in this research are explained, while section 4 presents the results and discussion of this work. The conclusions and outlook are provided in section 5.

## 2   Study area and hydrometeorological datasets

This study was tested in a semi-humid region in the Baden-Württemberg state of Germany (Figure 1) that characterized by temperate monsoon climate. Elevations of this state range from 85 m to 1 493 m above sea level. The heterogeneity of climate characteristics is mainly due to the great variability of elevations within the study area. Winters are mild whereas summers are warmer. The annual mean air temperature in Baden-Württemberg is about 10.2 °C. Precipitation is evenly distributed through the year. However, its seasonality shows a weak trend. The monthly rainfall reaches its peak in June, whereas the month of

October shows the least precipitation amount.

The meteorological observations used in this study was provided by the German Weather Service (DWD). Daily air temperature required for the rainfall-runoff model was interpolated on a $1 \times 1$ km$^2$ grid from the observations using the algorithm of External Drift Kriging (Ahmed and De Marsily, 1987). The topographical elevation was taken as external drift (Hundecha and Bárdossy, 2004; Das et al., 2008). The long term monthly potential evapotranspiration and the average air temperature were

used to compute the daily potential evapotranspiration using the Hargreaves and Samani method (Hargreaves and Samani, 1985).

Precipitation data from a dense network of daily precipitation stations (62 km$^2$/station in 1991) and from a less dense network of pluviometers (144 km$^2$/station in 1991) with high resolution precipitation observations were used for this study. All



available data from the time period 1991-2010 was considered. The number of available daily stations and pluviometers varies according to different time period. Figure 2 illustrates the number of available observation locations in Baden-Württemberg between the years 1991 and 2010. It can be seen from the graph, more than 430 daily stations were available in 1991, while only 30 pluviometers. The total number of daily stations decreased dramatically to 250 around 2003 and remained constant for the subsequent years. The number of pluviometers kept increasing throughout the whole period and experienced a sharp increase from 100 to 200 in the year 2005.

The following different precipitation datasets were created according to the available observed data:

1. High resolution observed precipitation was aggregated to hourly time steps and interpolated subsequently to a $1 \times 1$ km$^2$ grid using the ordinary Kriging (Matheron, 1963). The correlation function obtained from the cross-correlations of the hourly time series was used as a basis for the variogram. This set will be referred as Sparse Hourly (SH) set.

2. Observed daily precipitation combined with the daily aggregations of the high temporal resolution data were used to create a $1 \times 1$ km$^2$ gridded datasets using the ordinary Kriging. The variogram was based on the cross-correlations of the daily time series. This set will be referred as Dense Daily (DD) set.

3. High resolution precipitation was aggregated to daily time steps and interpolated subsequently for a $1 \times 1$ km$^2$ grid using the ordinary Kriging. The variogram was based on the cross-correlations of the aggregated daily time series. This set will be referred as Sparse Daily (SD) set.

4. Observed daily precipitation combined with the hourly aggregations of the high temporal resolution data were used to create a $1 \times 1$ km$^2$ grid using the disaggregation method rescaled ordinary Kriging (Bárdossy and Pegram, 2016). The variogram was based on the cross-correlations of the hourly time series. This set is denoted as Dense Hourly (DH) set.

Figure 3 illustrates the frame of these four different datasets. The DD and SD sets are practically the daily aggregations of the DH and SH sets. Note that DH is a dataset combining hourly observations and artificially disaggregated daily data. One of the research questions raised here is to find out if a disaggregation leads to an improvement of model performance. Comparisons of the model performances on the pairs of (SD, SH) and (DD, DH) provide information on the effect of temporal resolution. While comparisons between (SD, DD) and (SH, DH), provide information on the influence of the rainfall observation network density.

Four mesoscale catchments (Figure 1), namely Rottweil, Schwaibach, Pforzheim and Kocherstetten, were selected from the upstream region for testing the sensitivity of model performance to different rainfall datasets as described previously. The daily streamflow record of these catchments was collected for the period 1991- 2010. The basic characteristics for the study catchments are listed in Table 1.



## 3 Model and methodology

### 3.1 Model structure

The conceptual HBV model was introduced in the 1970s at the Swedish Meteorological and Hydrological Institute (SMHI) (Bergström and Forsman, 1973). Due to its simplicity, low demand of inputs and few model parameters, HBV model has been a

5 preferred model for rainfall-runoff simulation and flood forecasting. Figure 4 represents the structure diagram of HBV model (Singh, 2010). In general, three main modules are included in HBV model, namely snow routine, soil moisture routine and runoff routine (Hartmann, 2007; Singh, 2010).

First of all, the snow accumulation and melt process is estimated by the relatively simple degree-day method (Rango and Martinec, 1995) using two parameters: degree day factor ($DD$) and threshold temperature for snow/rain ($TT$) (as shown in

Equation 1). In this method, the measured precipitation is supposed to be solid (snowfall) if the air temperature is lower than the threshold temperature, otherwise, precipitation appears liquid state (rainfall) if the weather is warmer than the threshold value.

$$Snowmelt = DD \cdot (T - TT), \quad \text{if} \quad T > TT \tag{1}$$

In HBV model, soil moisture storage is decided by balancing rainfall and evapotranspiration according to two soil moisture

constants: permanent wilting point ($PWP$) and field capacity ($FC$). The soil wetness index, which is represented by the ratio of direct runoff to effective precipitation ($\frac{\Delta Q}{\Delta P}$) can be estimated by:

$$\frac{\Delta Q}{\Delta P} = (\frac{SM}{FC})^{Beta} \tag{2}$$

where $SM$ denotes the actual soil moisture and $Beta$ determines the proportion of effective precipitation contributing to runoff at a given soil moisture state. The approach of Penman equation is used to estimate the potential evapotranspiration according

to the long-term monthly mean air temperature ($T_M$) and long-term monthly average potential evapotranspiration ($PE_M$) (Penman, 1948):

$$E_{tp} = (1 + C(T - T_M))PE_M \tag{3}$$

Here $C$ is the evapotranspiration coefficient. The actual evapotranspiration ($E_{ta}$) can be estimated as follow:

$$E_{ta} = \begin{cases} E_{tp} & \text{if} \quad SM > PWP \\ \frac{SM}{PWP} \cdot E_{tp} & \text{else} \end{cases} \tag{4}$$

As shown in Equation 2, runoff response routine is calculated by a non-linear function based on excessive effective precipitation and actual soil moisture. The runoff concentration process consists upper and lower reservoirs with five free parameters:

$$Q_0 = K_0(S_1 - HL) \tag{5}$$





$$Q_1 = K_1 S_1 \tag{6}$$

$$Q_d = K_d S_1 \tag{7}$$

$$Q_2 = K_2 S_2 \tag{8}$$

The runoff is divided into surface flow ($Q_0$), interflow ($Q_1$) and baseflow ($Q_2$) with three recession coefficients $K_0$, $K_1$ and $K_2$, along with a conceptual threshold water level (*HL*) for generating surface flow. The two parallel reservoirs are connected in the form of percolation storage ($Q_d$) from upper reservoir to the lower one with the parameter of percolation constant $K_d$.

Finally, a transformation function approach with the triangular weighting parameter *MAXBAS* is used to smooth the generated total runoff ($Q_0 + Q_1 + Q_2$) to obtain discharge at the outlet.

In this study, for investigating the sensitivity of model performance on the spatial resolution of input variables, two HBV models with different levels of complexity were applied: lumped HBV and spatially distributed HBV, respectively. In the lumped model, precipitation, temperature and potential evapotranspiration were supposed to be equally distributed among the

catchment and all the processes were calculated for the whole catchment. Previous studies have indicated that the altitude is an important reason for the spatial differentiation of meteorological elements, such as temperature, precipitation, evapotranspiration and snow melt. Therefore, the spatially distributed HBV model was constructed to separated the whole catchment into several zones based on topographic elevation. The $1 \times 1$ km$^2$ grid based precipitation and temperature data were computed averagely according to elevation zone and used as inputs for model simulation. In the spatially distributed model, the snowmelt

and soil moisture modules related parameters can be adjusted differently for each elevation zone. The parameters controlling the runoff response processes were estimated for the whole catchment similarly to the lumped model (Das et al., 2008).

There are 15 parameters describing the HBV model, where only 9 parameters were selected for calibration in this study. Table 2 lists the  initial upper and lower limit of  parameters that will be optimized by model calibration using historical data. The data depth based parameter optimization method-Robust Parameter Estimation (ROPE) algorithm (Bárdossy and Singh,

2008) was applied for model parameter identification. The ROPE approach could lead to a certain number of model parameters with ideal model performance (Bárdossy et al., 2016). For this study, each simulation results in 10 000 heterogeneous parameter sets with similar and good model performance.

### 3.2 Performance criteria

In this study, the Nash-Sutcliffe (NS) efficiency coefficient (Nash and Sutcliffe, 1970) between the observed and simulated

streamflow was used to access the model performance:

$$NS = 1 - \frac{\sum_{t=1}^{T} \left( Q_o(t) - Q_m(t) \right)^2}{\sum_{t=1}^{T} \left( Q_o(t) - \bar{Q}_o \right)^2} \tag{9}$$



where $Q_o(t)$ and $Q_m(t)$ are the observed and simulated discharges respectively and $\bar{Q}_o$ is the mean of observed discharge series.

Meanwhile, the Mean Square Error (MSE) of the flow for the time period that the observed discharge is greater than or equal to 90% high flow value was calculated to assess the flood forecasting ability of the models:

$$MSE = \frac{1}{n}\sum_{i=1}^{n}(Q_0(i) - Q_m(i))^2 \tag{10}$$

Here $Q_o(i)$ and $Q_m(i)$ are the observed and modeled discharges when the observed discharge is greater than or equal to 90% high flow value.

### 3.3 Model calibration experiments

A split sample calibration methodology has been applied in this study to separate the whole data series into two equal periods: 1991-2000 and 2001-2010. Model calibration was carried out for both time periods and a cross-validation analysis was performed as well. For each calibration run, the first water year data was used as warm-up period to reduce initial errors and was not used to evaluate the model performance.

In this study we investigated the impacts of using different methods for spatial interpolation of hourly rainfall data on model performance. The four rainfall datasets were assigned as input variables for model calibration and validation. We also assessed the effects of the temporal and spatial resolutions of the inputs on the model performance in terms of Nash-Sutcliffe efficiency and the mean square error of the high flow. We conducted experiments of model calibration for a lumped and a spatially distributed HBV model using hourly and daily input variables, respectively. For the spatially distributed model structure, a contour interval of 100 m was taken to divide the whole study catchment into several elevation zones. Note that all the model calibrations were performed on the basis of simulating daily discharge. Due to the limitation of observed temperature, air temperature and potential evapotranspiration were assumed to be constant over the whole day.

We also wonder if the combination of daily scale model and hourly scale model leads to a better prediction in streamflow. It is interesting to investigate the similarities of different temporal resolution. Therefore, the common calibration tragedy was proposed in this study to calibrate the daily scale model and hourly scale model simultaneously. This kind of approach is expected to identify robust model parameters for the application of model in different temporal resolutions. Common calibration approach is a multi-objective optimization function and the compromise programming method (Zeleny, 1981) was used to formulate the objective function:

$$O(\theta) = \sum_{i=1}^{n}(NS_i^* - NS_i(\theta))^p \tag{11}$$

Here index $i$ indicates the type of temporal resolution, $NS_i^*$ means the optimal model performance which can be represented by the individual calibrated model performance. Here we aim to minimize the value of objective function $O(\theta)$. For the balancing factor $p$, a moderately high $p = 4$ was given in this study. More details about the common calibration of hydrological models strategy can be found in Bárdossy et al. (2016).



## 4 Results and discussion

### 4.1 Comparison of the rainfall datasets

Firstly, the quality of the rainfall products was assessed and compared for four selected catchments. As the SD and DD sets are the daily aggregations of the SH and DH sets, here we only compared the daily precipitation sets SD and DD for both calibration decades (Figure 5). It can be seen clearly from the figures that the interpolated precipitation datasets display some difference for all study catchments. The asymmetry of the scatterplots is fairly obvious for the first decade (1991-2000). In general, the DD dataset leads to higher value than the SD dataset. The reason behind this is mainly because the low density of pluviometers observations during the period of 1991-2000 leads to big errors in the spatial interpolation of rainfall. This is especially the case for Schwaibach catchment which varies strongly in geographical elevation (from 190 m a.s.l. to 1028 m a.s.l.). For the period 2001-2010, the SD and DD sets become similar in magnitude along with the increasing of available sub-daily observations.

### 4.2 Calibration and validation model performance

As designed in section 3.3, for the selected catchments, model calibrations were carried out using four rainfall datasets for both lumped and spatially distributed HBV models. Data series from 1991 to 2010 were average split into two sub-periods for calibration and cross-validation. This leads to 16 calibration runs and 16 validation runs for every catchment. As mention before, each simulation could obtain10 000 parameter sets with similar model performance. To make it simple, we took the mean value of the corresponding 10 000 model performances to represent the model efficiency.

Table 3 lists the average value of the NS model performance for the four selected catchments using lumped HBV model and Table 4 lists the simulated NS performance for spatially distributed version of the model, respectively. The results show that all four datasets can reproduce relatively accurate historical daily streamflow series for all selected catchments. Results also indicate that the model performances vary across catchments. The Kocherstetten catchment generally performs the best with an average NS value of 0.84 for all simulations, while the Pforzheim catchment has the worst mean NS performance of 0.58 for all calibration runs. Moreover, for a specific catchment, the calibrated model performances for different data periods are also different. For the Schwaibach and Pforzheim catchments, the calibrated model performance for the time period 2001-2010 is obviously better than the performance for the time period 1991-2000 for most of the datasets. This might be due to the increasing of the raingauge density inside or nearby the catchment and the quality of rainfall data with the development of time and technological progress. In particular, the model calibrations for the period 1991-2000 of the Schwaibach catchment using the sets SH and SD perform very weak for both calibration and validation; the loss in NS coefficient is about 0.3 when compared to the corresponding results of the sets DH and DD. This indicates that systematic interpolated precipitation errors have critical influence on model calibration.

The flexibility of model in flood prediction is analyzed with the behavior of high flow. Tables 5 and 6 list the mean square error of 90% high flow for lumped model and spatially distributed model, respectively. Figure 6 shows the flow duration curve for the natural logarithm of simulated and observed discharge for all study catchment for the years between 2001 and 2010.





Results indicate obviously that for most of the calibration runs, the set DH performs the best for the high flow, followed by set SH, set DD performs a little weaker than set SH, while set SD has the worst performance in the flood simulation.

## 4.3 Comparison of the performance corresponding to the temporal resolution

Firstly, the model performance of different temporal resolution was compared for all datasets and model structures. For the pairwise comparison, all the conditions are the same in the model except for the temporal resolution of input variables (hourly and daily). The results of the sparse sets and dense sets are separated here. Figure 7 compares the model performance of using hourly and daily rainfall variables as model input for the precipitation sets that were interpolated using only high-resolution precipitation observations (SH, SD). Figure 8 compares the corresponding results for the rainfall datasets that incorporated observed daily value with high-resolution observations (DH, DD). The result shows that all the scatters are laying below the diagonal for the different level of observation density. For both calibration and validation periods, the simulations using hourly data as model input outperform the one that based on the daily resolution. For the dataset with low observation network density, the average NS value of set SH is about 0.73 for calibration period and 0.68 for validation period, while the mean NS coefficient that was calibrated using SD set is 0.67 and 0.6, respectively. The higher observation density datasets show a similar tendency. The mean NS value of using DH set is around 0.79 for calibration and 0.77 for validation, while the result of set DD is 0.72 and 0.69, respectively. The hourly scale model performs better than the daily model indicating that the dynamic runoff of catchment could be better simulated with a higher temporal resolution of input variables. According to the distances from the diagonal to the scatterplots, we could find that the difference in model performance for different temporal resolution is larger for the catchments with relatively low NS model performance, such as Schwaibach and Pforzheim. For Rottweil and Kocherstetten, the model performance of hourly calibrated model is only slightly better than the daily based model.

## 4.4 Comparison of the performance corresponding to observation density

Results also indicate that rainfall data network density has significant impact on model simulation and parameter optimization. Figure 9 plots the simulated NS coefficient for the daily datasets that was interpolated using different density of rainfall observation network. It shows obviously from the location of points that the simulated model performance of DD set is generally better than the result of SD set for both calibration and validation periods. The average NS model performance of DD set over all simulations is about 0.71 while the value for SD set is 0.64. The model performance for the hourly based simulation shows similar trend as the model performance for the daily time step. As shown in Figure 10, the model calibration of DH set outperform the result of SH set. The results demonstrate that the high observation density could lead to considerable improvement of model performance for both daily and hourly time scales.

Figure 11 illustrates the cumulative distribution function of NS model performance using sets SD, SH and DH for model calibration (left) and validation (right). As can be seen clearly from the curves, if precipitation data comes from a sparse network of pluviometers, higher temporal resolution datasets (as represented by set SH) can achieve better model performance than the lower ones (as represented by set SD). Decreasing the length of time step in model simulation could provide a better fit of daily





streamflow. In addition, the combination of observed high-resolution observations with disaggregated daily precipitation (as represented by set DH) leads to a further improvement of daily streamflow prediction.

## 4.5 Comparison of the performance corresponding to the spatial resolution

The performance of different model structures in terms of different spatial resolutions was assessed by comparing performance for lumped HBV model and spatially distributed HBV model. Figure 12 compares the NS model performance for these two model structures for calibration (left) and validation (right) periods. The correlation between model performance and the spatial resolution of model seems not clear for the study catchments. For some simulations, the elevation zone based spatially distributed models outperform the lumped ones, especially for the catchments having high NS coefficient. Despite the increase in model performance being only marginal. However, for the catchments with relatively weak model performance, the lumped model could even lead to slightly better performance than the semi-distributed model structure, especially for the validation period that the difference seems larger than the calibration period. It indicates that for model validation, the model parameters estimated by distributed HBV model shows weaker transferability. Possible explanation for this case could be that the distributed model structure raises the number of parameters to be identified and the parameters are underestimated during the calibration period. We can conclude from this comparison that the improvement in spatial resolution of model structure did not clearly enhance the model performance. However, it is surprising since we expected a better model performance with a higher spatial resolution of model and a complicated set of parameters. The results support the findings of Das et al. (2008) that distributed model structures does not significantly improve model performance.

The complex structure version of model did not perform better than the lumped model incurrent research. This might be due to the lack of underlying surface information and the calibration procedure was not enough for the identification of distributed model parameters. A second explanation could be that the temporal resolution of the force inputs is not sufficient for distributed model structure.

## 4.6 Common calibration of models with different temporal resolutions

As shown before, the combination of hourly observations and daily observations lead to the improvement of data quality as the sets DH and DD show better model performance than the sets SH and SD. Furthermore, common calibration of lumped HBV model was performed for the sets DH and DD to identify model parameters good for both hourly and daily time steps. It is important to note that the value of time step dependent parameters ($DD$, $K_0$, $K_1$, $K_d$ and $K_2$) should be converted according to the temporal resolution of model. The common calibration was performed for two decades separately, and the cross-validation analysis was performed as well. The common calibration and validation results were compared with the individual calibration cases (Figure 13). For the calibration period, the common calibration always leads to slightly weaker performance for all datasets. For three of the DD datasets, model performances of common parameters are rather similar to individual calibration results. The average loss of NS model performance over all catchments is about 0.02 for set DH and 0.01 for set DD. For the validation period, from the scatterplots, it is clearly seen that the common parameters outperform the individual ones for about half of the all simulations. It indicates that common calibrated parameters based on different time steps could be a feasible





approach for increasing the temporal transferability of models. The reason for the robustness of common parameters might be that common calibration tragedy could provide more information for identifying model parameters.

The calibrated model parameters using daily precipitation, hourly precipitation and common calibration tragedy were also compared in this study. Figure 14 and Figure 15 show the distribution of the optimized model parameters for Rottweil and Pforzheim, respectively. Note that all the parameter values have been normalized by the initial range that listed in Table 2. Form the box plots we could find that some model parameters, especially the shape factor ($Beta$) and the threshold water level for surface runoff ($L$), strongly depend on the selected precipitation dataset.

## 5    Conclusions and outlook

This paper investigated the impact of the observed precipitation data in model simulation and parameter estimation. The sensitively of model performance to different temporal and spatial resolutions of input variables were also tested in this study. Two different model structure, lumped and spatially distributed HBV models, were used to simulate daily runoff using precipitation data sets with different time resolutions and interpolated using different observation network density. The models were applied to four upstream catchments using NS coefficient as objective function and the mean square error for the high flow was also assessed. The common calibration scenario was proposed to calibrate the model with different time scale simultaneously to provide robust model parameters.

The calibration results indicate that rainfall data quality has a significant impact on model performance. Interpolation of hourly precipitation using disaggregated daily value as additional information could potentially enhances the quality of the data and reduces the uncertainty of the model inputs. The result shows that higher temporal resolution could significantly improved the model performance if the observation density was high. A combination of observed high-resolution observations with disaggregated daily precipitation leads to a further improvement. For the present study, the lumped and spatial distributed model structures perform very similar indicating that higher model resolution does not or only marginally improve the model performance.

A great amounts of efforts had been made to improve the performance of rainfall-runoff model in recent years. The results of this study suggested that higher temporal resolution of inputs always outperform the lower ones, an effective data disaggregation could lead to an improvement of the model performance. Results also indicated that higher spatial resolution of model, which cause the complexity of model structure and parameters, do not always enhance the model performance. Compared with the idea of increasing the spatial resolution by distributed model, increasing the temporal resolution of model inputs by disaggregation method could be an easier and much lower cost way to improve model performance.

In this study, all the hourly model outputs were aggregated into daily and only the daily streamflow was involved in the evaluation of model performance. As the daily-based rainfall-runoff response of a catchment is mostly dominated by rainfall amount and actual evapotranspiration, we believe that the variability in precipitation have rather strong impacts on the smaller temporal scales, such as the hourly response of discharge. Meanwhile, in the spatially distributed model, the subcatchments were separated only based on the topographic elevation, which might be not enough to represent the full spatial variability.

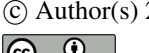


Moreover, the impacts of precipitation to the hourly response and the full distributed model structure could be considered as the next phase of work.

*Acknowledgements.* This study was partially supported by the National Key Research and Development Program of China (2016YFC0402701), National Science Foundation of China (51879067), Fundamental Research Funds for the Central Universities of China (2015B28514), the

5  China Postdoctoral Science Foundation(2017M621614), and the Priority Academic Program Development of Jiangsu Higher Education Institutions. Special thanks to Gebdang Biangbalbe Ruben for proofreading of the manuscript.



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





**Table 1.** Catchment characteristics for the 4 selected catchments.

| No. | Streamgauge name | Longitude ($^o$E) | Latitude ($^o$N) | Area (km$^2$) | Elevation (m) | Annual precipitation (mm) | Average temperature ($^o$C) | Annual runoff (mm) |
|---|---|---|---|---|---|---|---|---|
| 1 | Rottweil, Neck | 8.38 | 48.10 | 455 | 555-1010 | 929.0 | 9.7 | 363.2 |
| 2 | Schwaibach,Kinzig | 8.02 | 48.24 | 955 | 190-1028 | 1331.8 | 9.7 | 757.3 |
| 3 | Pforzheim,Würm | 8.43 | 48.52 | 417 | 357-583 | 761.7 | 9.3 | 232.9 |
| 4 | Kocherstetten, Kocher | 9.45 | 49.16 | 1288 | 292-698 | 930.6 | 9.4 | 401.6 |





**Table 2.** Description of HBV model parameters and parameter ranges for model calibration.

| Parameter | Description | Max | Min |
|-----------|-------------|-----|-----|
| TT | Threshold temperature for snow melt initiation ($^0$C) | 2 | -2 |
| DD | Degree-day factor | 3 | 1.5 |
| FC | Field capacity (mm) | 600 | 50 |
| Beta | Shape coefficient | 8 | 0.2 |
| HL | Threshold water level for near surface flow (mm) | 100 | 1 |
| $K_0$ | Near surface flow storage constant | 0.8 | 0.2 |
| $K_1$ | Interflow storage constant | 0.25 | 0.1 |
| $K_d$ | Percolation storage constant | 0.2 | 0.05 |
| $K_2$ | Baseflow storage constant | 0.1 | 0.01 |





**Table 3.** Average NS model performance for the lumped HBV model.

| Catchment | Precipitation data set | Calibration for 1991-2000 | Calibration for 2001-2010 | Validation for 1991-2000 | Validation for 2001-2010 |
|---|---|---|---|---|---|
| Rottweil | SH | 0.71 | 0.71 | 0.65 | 0.65 |
| | DH | 0.79 | 0.73 | 0.73 | 0.68 |
| | SD | 0.61 | 0.61 | 0.56 | 0.55 |
| | DD | 0.67 | 0.63 | 0.63 | 0.59 |
| Schwaibach | SH | 0.60 | 0.88 | 0.52 | 0.72 |
| | DH | 0.89 | 0.88 | 0.88 | 0.87 |
| | SD | 0.57 | 0.85 | 0.49 | 0.68 |
| | DD | 0.84 | 0.86 | 0.83 | 0.83 |
| Pforzheim | SH | 0.61 | 0.69 | 0.60 | 0.65 |
| | DH | 0.63 | 0.69 | 0.63 | 0.67 |
| | SD | 0.48 | 0.60 | 0.46 | 0.56 |
| | DD | 0.48 | 0.60 | 0.49 | 0.57 |
| Kocherstetten | SH | 0.88 | 0.85 | 0.86 | 0.84 |
| | DH | 0.89 | 0.85 | 0.87 | 0.84 |
| | SD | 0.84 | 0.84 | 0.81 | 0.79 |
| | DD | 0.84 | 0.83 | 0.81 | 0.81 |





**Table 4.** Average NS model performance for the distributed HBV model.

| Catchment | Precipitation data set | Calibration for 1991-2000 | Calibration for 2001-2010 | Validation for 1991-2000 | Validation for 2001-2010 |
|---|---|---|---|---|---|
| Rottweil | SH | 0.70 | 0.68 | 0.63 | 0.55 |
| | DH | 0.80 | 0.69 | 0.74 | 0.66 |
| | SD | 0.61 | 0.59 | 0.54 | 0.46 |
| | DD | 0.68 | 0.60 | 0.63 | 0.57 |
| Schwaibach | SH | 0.59 | 0.88 | 0.50 | 0.76 |
| | DH | 0.90 | 0.88 | 0.88 | 0.87 |
| | SD | 0.55 | 0.86 | 0.47 | 0.72 |
| | DD | 0.85 | 0.86 | 0.84 | 0.85 |
| Pforzheim | SH | 0.55 | 0.68 | 0.55 | 0.64 |
| | DH | 0.59 | 0.67 | 0.59 | 0.64 |
| | SD | 0.42 | 0.58 | 0.41 | 0.54 |
| | DD | 0.45 | 0.58 | 0.46 | 0.54 |
| Kocherstetten | SH | 0.88 | 0.86 | 0.86 | 0.84 |
| | DH | 0.89 | 0.86 | 0.87 | 0.84 |
| | SD | 0.84 | 0.84 | 0.82 | 0.80 |
| | DD | 0.84 | 0.84 | 0.82 | 0.81 |





**Table 5.** Mean square error of the 90% high flow for the lumped HBV model.

| Catchment | Precipitation data set | Calibration for 1991-2000 | Calibration for 2001-2010 | Validation for 1991-2000 | Validation for 2001-2010 |
|---|---|---|---|---|---|
| Rottweil | SH | 83.1 | 74.6 | 118.7 | 83.5 |
| | DH | 55.1 | 69.8 | 82.4 | 84.7 |
| | SD | 120.0 | 104.5 | 151.4 | 108.5 |
| | DD | 101.7 | 98.9 | 120.0 | 110.1 |
| Schwaibach | SH | 2511.4 | 338.6 | 3214.9 | 663.6 |
| | DH | 565.4 | 324.4 | 722.7 | 328.2 |
| | SD | 2739.9 | 401.1 | 3423.0 | 805.7 |
| | DD | 916.0 | 389.2 | 1048.1 | 448.2 |
| Pforzheim | SH | 11.8 | 7.3 | 12.4 | 8.3 |
| | DH | 11.2 | 6.9 | 11.8 | 7.3 |
| | SD | 19.1 | 10.6 | 19.6 | 12.0 |
| | DD | 18.9 | 10.3 | 19.5 | 10.9 |
| Kocherstetten | SH | 438.9 | 457.5 | 545.5 | 558.7 |
| | DH | 288.5 | 439.3 | 350.5 | 518.8 |
| | SD | 651.9 | 551.9 | 801.9 | 760.4 |
| | DD | 556.0 | 544.1 | 665.0 | 701.3 |





**Table 6.** Mean square error of the 90% high flow for the distributed HBV model.

| Catchment | Precipitation data set | Calibration for 1991-2000 | Calibration for 2001-2010 | Validation for 1991-2000 | Validation for 2001-2010 |
|---|---|---|---|---|---|
| Rottweil | SH | 89.0 | 86.8 | 127.8 | 120.1 |
| | DH | 56.5 | 85.2 | 80.1 | 95.0 |
| | SD | 121.0 | 113.6 | 161.4 | 144.5 |
| | DD | 100.6 | 111.5 | 119.6 | 121.9 |
| Schwaibach | SH | 2657.1 | 326.9 | 3330.8 | 527.1 |
| | DH | 526.1 | 311.4 | 680.7 | 317.7 |
| | SD | 2869.6 | 387.9 | 3546.7 | 681.5 |
| | DD | 892.8 | 376.5 | 983.2 | 405.9 |
| Pforzheim | SH | 12.5 | 7.1 | 12.7 | 8.1 |
| | DH | 11.9 | 6.7 | 12.4 | 7.2 |
| | SD | 19.6 | 10.3 | 19.7 | 11.5 |
| | DD | 19.5 | 9.9 | 19.6 | 10.6 |
| Kocherstetten | SH | 425.7 | 455.1 | 541.2 | 551.5 |
| | DH | 293.5 | 429.1 | 355.3 | 515.1 |
| | SD | 633.3 | 552.0 | 778.6 | 727.3 |
| | DD | 542.4 | 540.8 | 637.0 | 670.9 |





**Figure 1.** Locations of the pluviometers(hourly) and daily rain gauges in Baden-Württemberg and the four selected catchments.





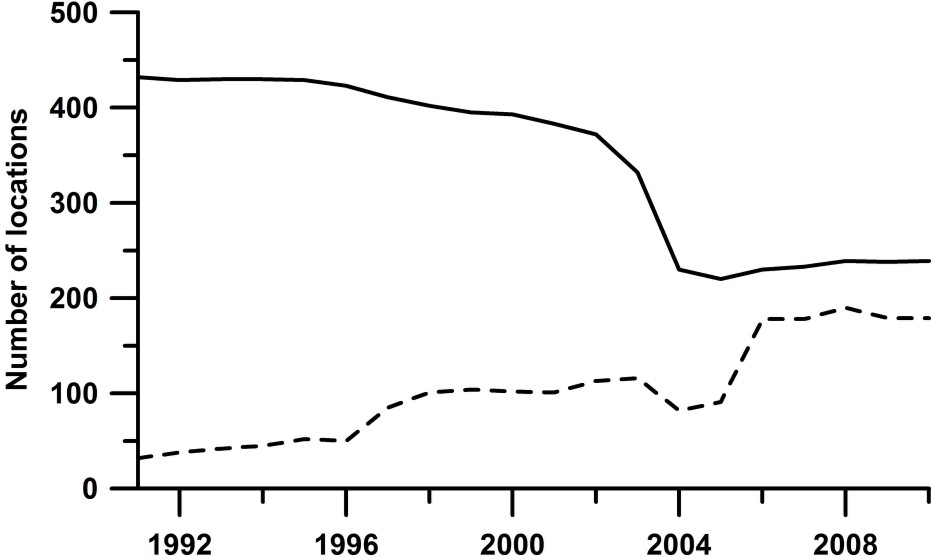

**Figure 2.** The number of available observation locations. Daily stations - solid line, pluviometers - dashed line.





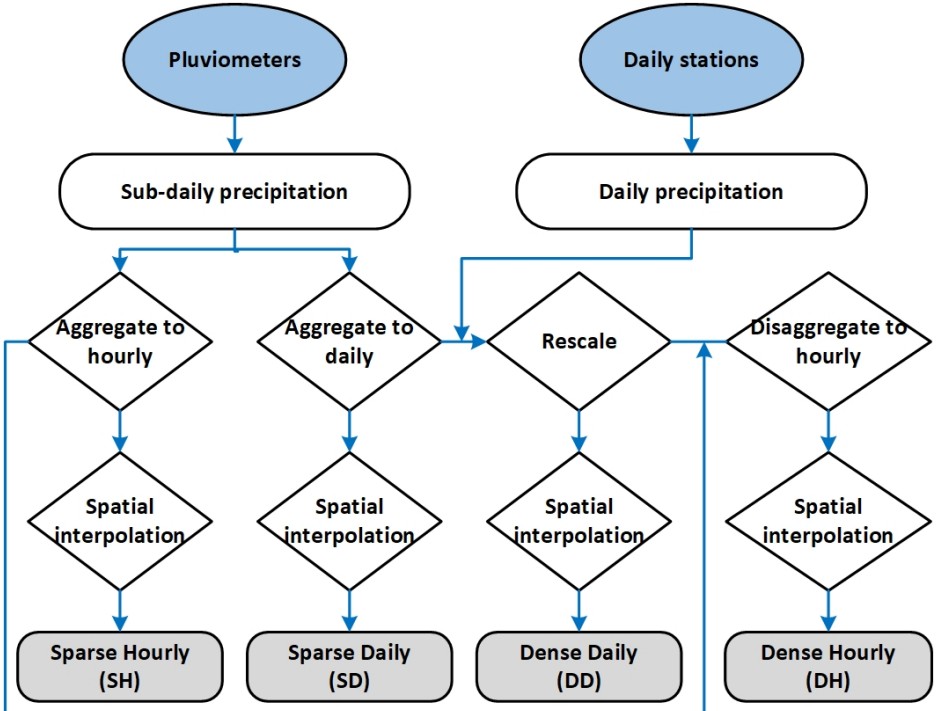

**Figure 3.** Schematic representation of four different precipitation data sets.




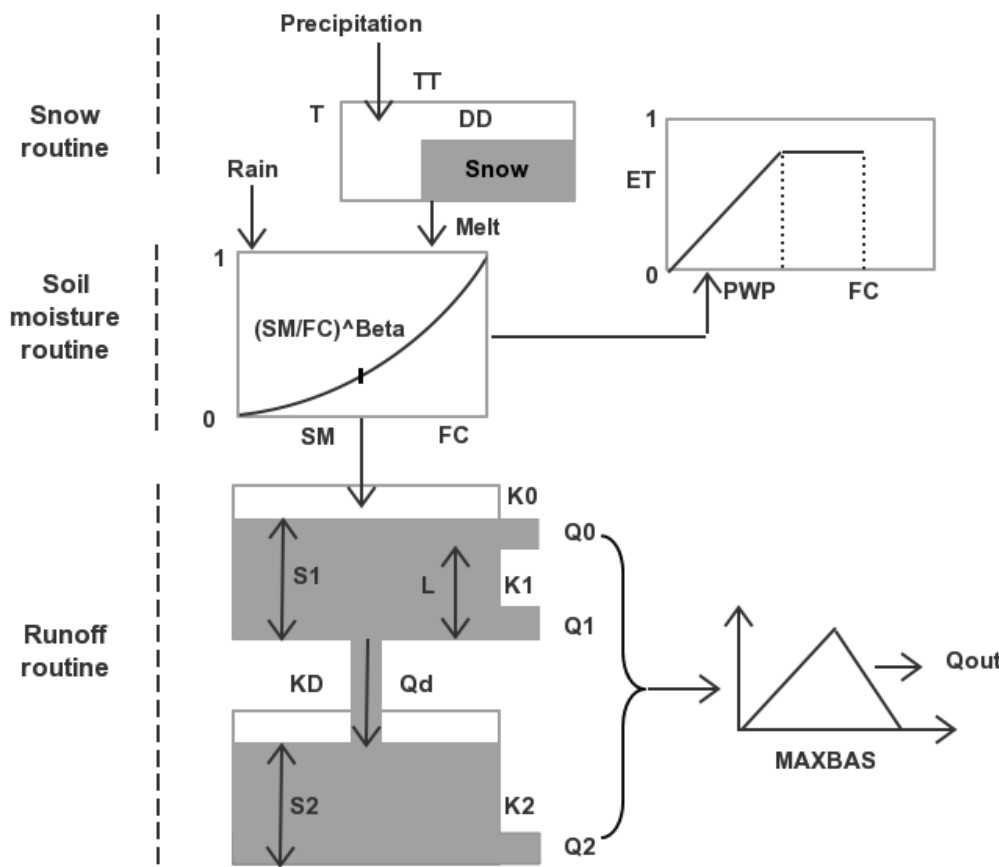

**Figure 4.** Schematic representation of HBV model.



**Figure 5.** Comparison of the daily precipitation data that interpolated using different observation network density.



**Figure 6.** Comparison of the simulated flow duration curve.





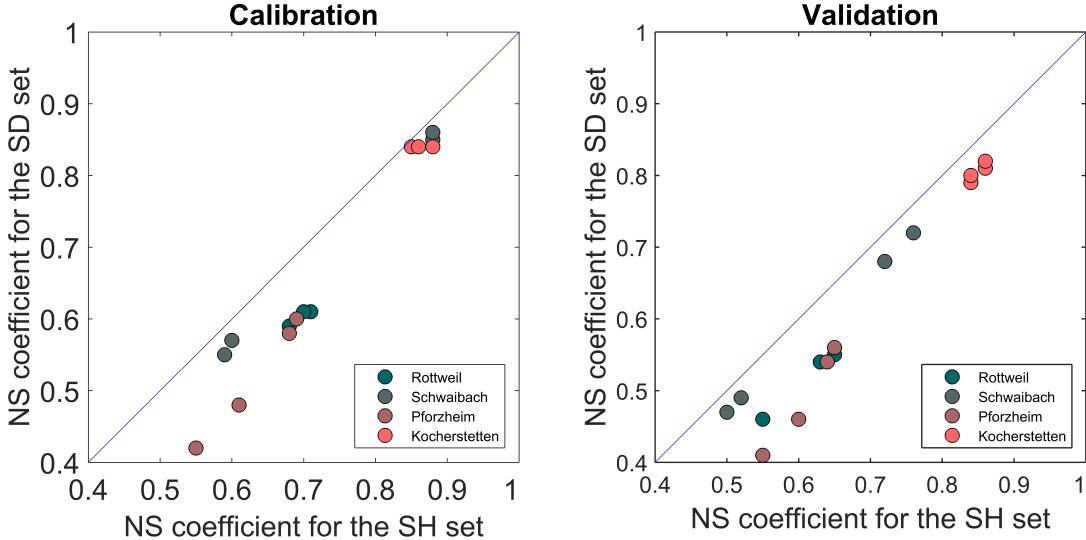

**Figure 7.** Comparison of NS model performance for using hourly and daily variables as model input for the SH and SD sets.





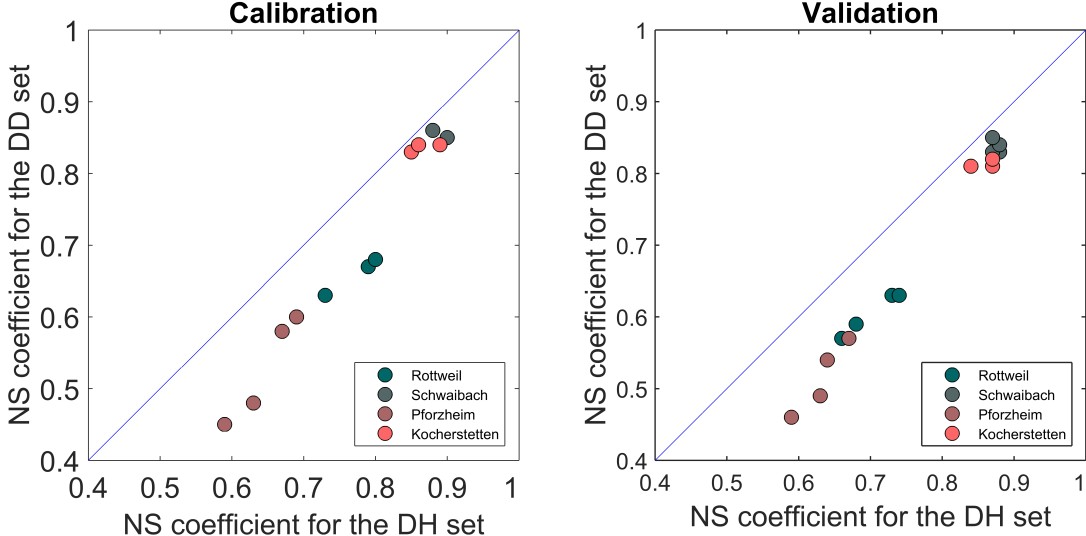

**Figure 8.** Comparison of NS model performance for using hourly and daily variables as model input for the DH and DD sets.





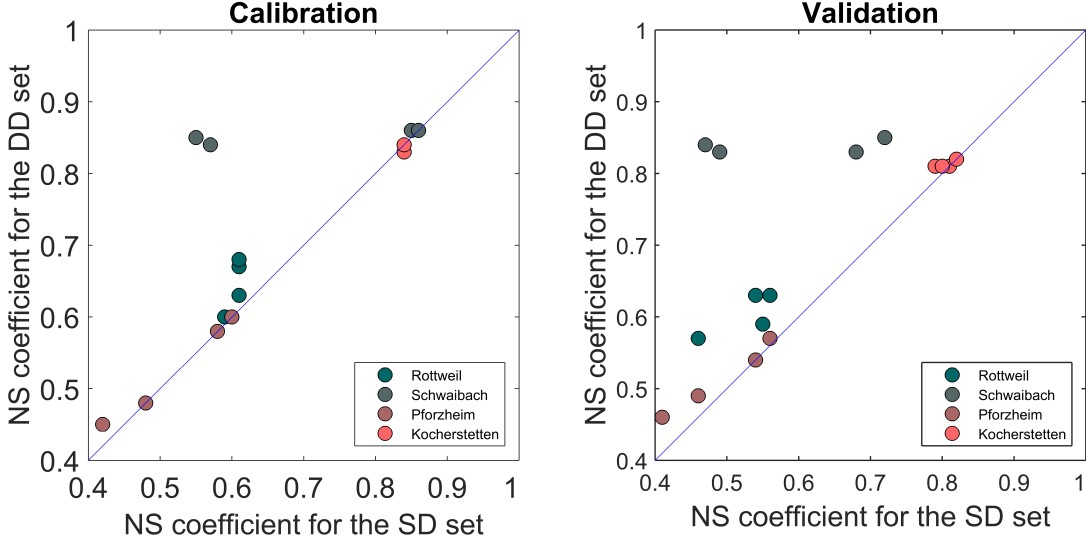

**Figure 9.** Comparison of model performance for different density of rainfall observation network, models were simulated based on daily time step.




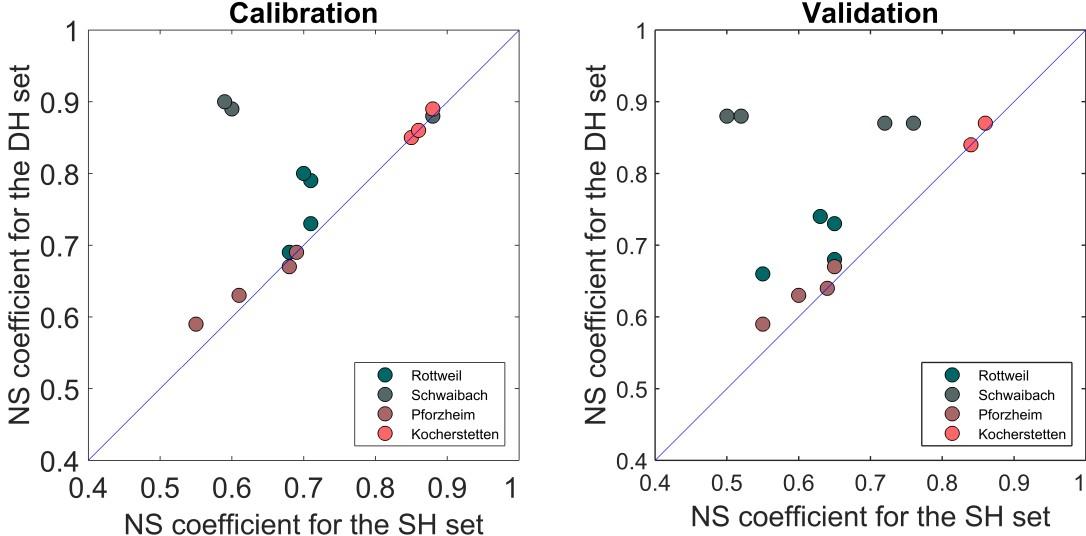

**Figure 10.** Comparison of model performance for different density of rainfall observation network, models were simulated based on hourly time step.





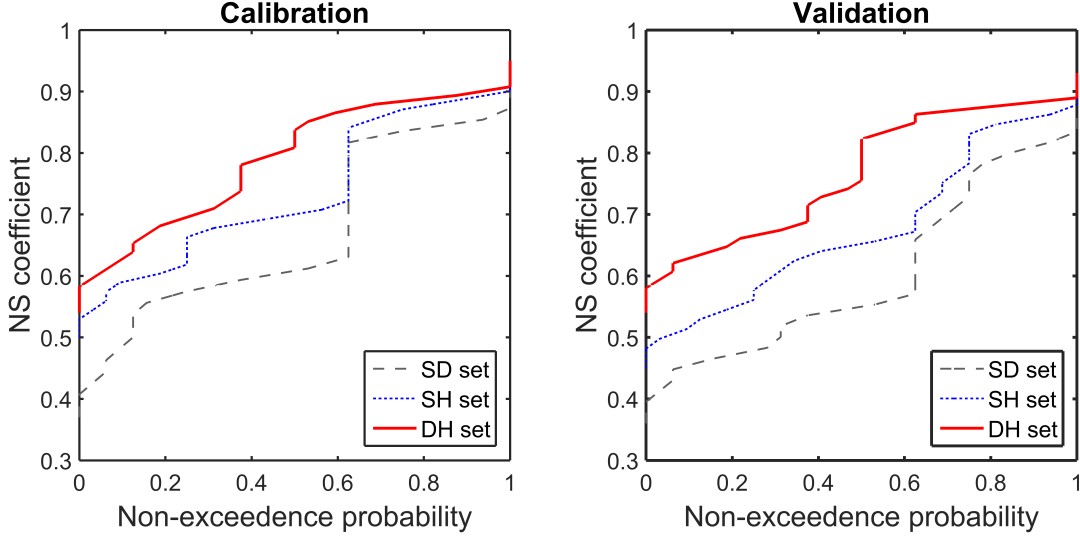

**Figure 11.** Cumulative distribution of NS coefficient for model calibration using different precipitation datasets .





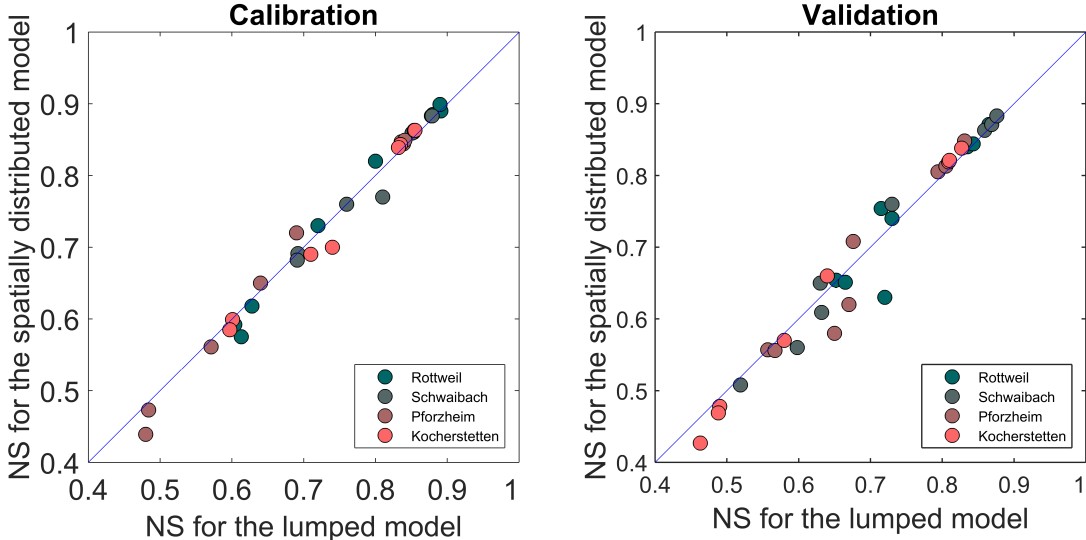

**Figure 12.** Comparison of model performance for different spatial resolution of model structure.





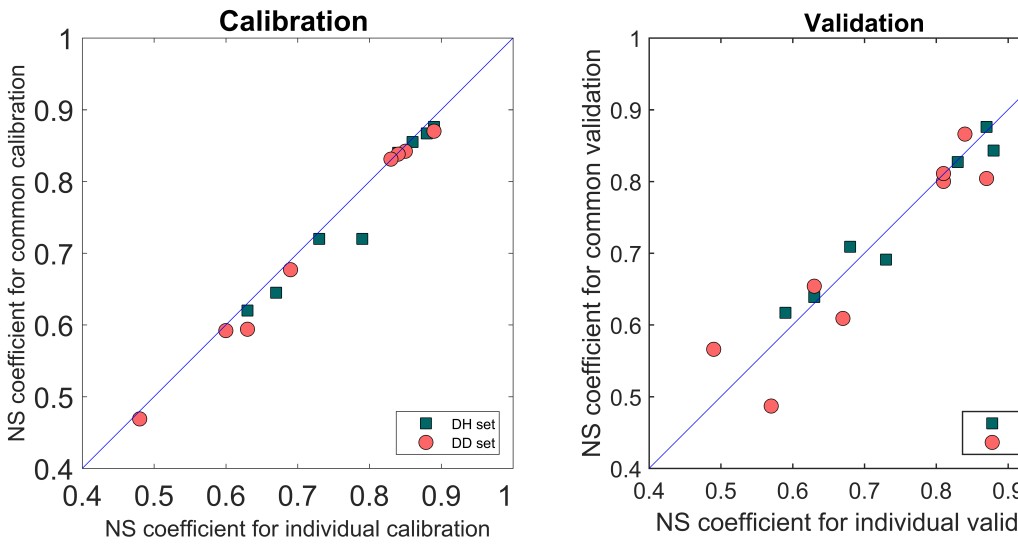

**Figure 13.** Comparison of model performance for individual calibration and common calibration for different temporal resolution datasets.




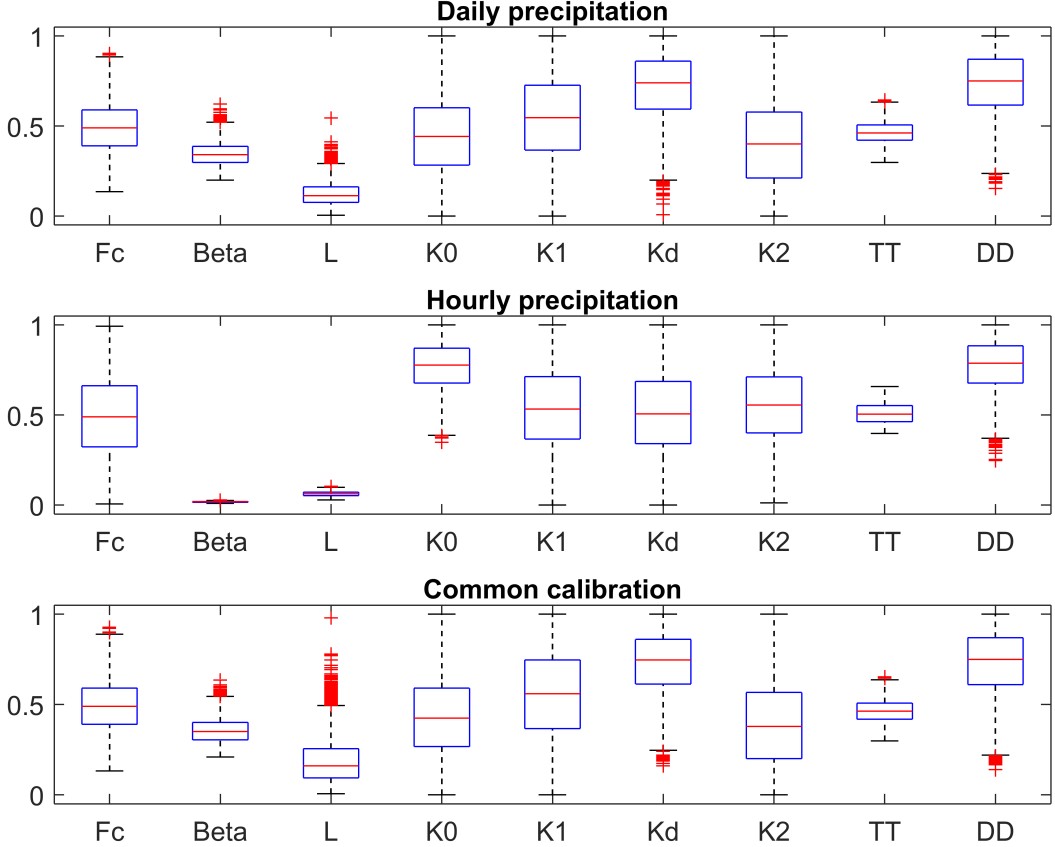

**Figure 14.** Comparison of model parameters for different temporal resolution for Rottweil catchment.





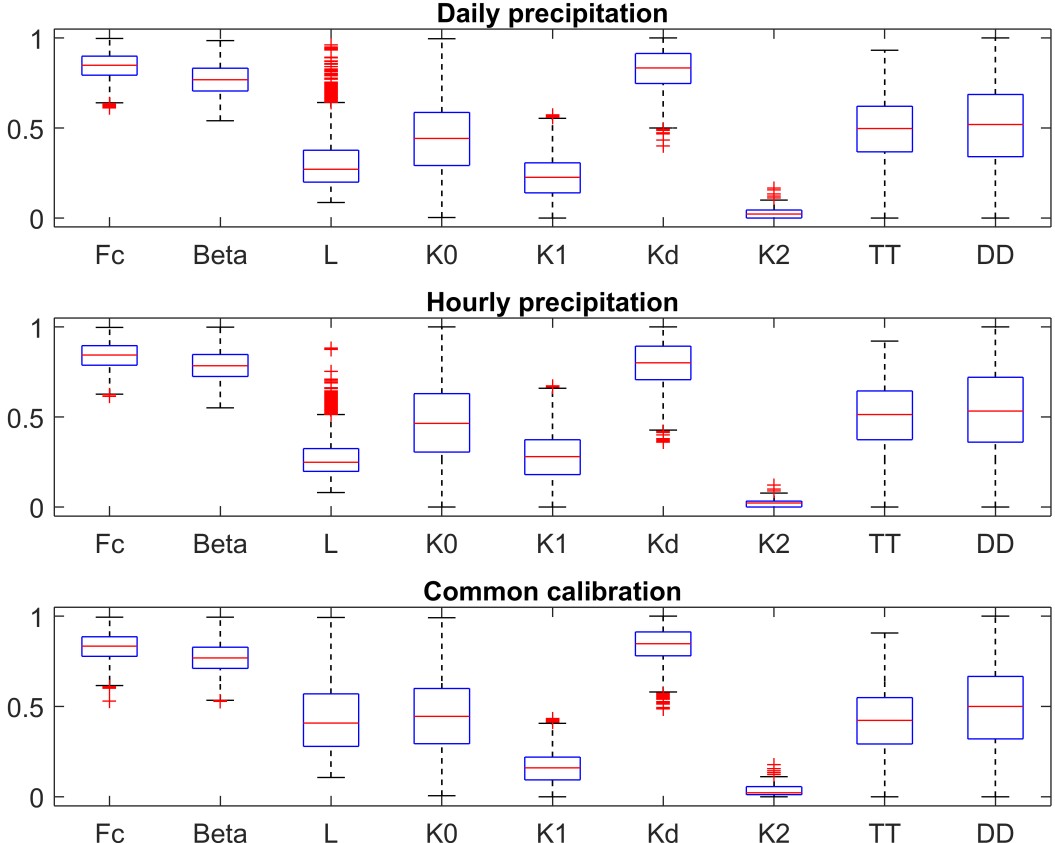

**Figure 15.** Comparison of model parameters for different temporal resolution for Pforzheim catchment.