# Peer review of "Sensitivity of hydrological model to the temporal and spatial resolutions of rainfall input"

_Hydrology and Earth System Sciences, 2018_

## Referee Comment (RC1) · Anonymous Referee #1 · 8 Dec 2018

This paper evaluates the sensitivity of a hydrological model to different temporal and spatial resolutions of rainfall input. The study uses four mesoscale catchments as a case study and evaluates the model's ability to capture NSE and MSE. The paper has a clear experimental design and identifies several interesting results.

The major limitation of the paper for me is the introduction and discussion. There are very little references to the wider literature and both sections do not address where this research sits in the wider field. Furthermore, the choice of catchments and performance criteria need to be better justified. My detailed comments are listed below.

Major Comments

[Figure]

Introduction – The introduction is quite short and I don't think gives the reader a thorough overview of previous literature on this topic and where this research sits within the field. There have been lots of other studies that have focused on the impacts of spatial and temporal resolution of rainfall on hydrological model output and you need to clearly explain how your research builds on these previous studies. I found it difficult to identify from the introduction what the research gap was and how this study addressed that research gap.

Study area and hydrometeorological datasets – The rationale for your choice of catchments needs to be outlined. Why were these four catchments chosen? Do they have different climatological characteristics that make them interestingly different? A lot of the following analysis focuses on differences between these mesoscale catchments so it is important that the reader understands what these key differences are. Table 1 contained some interesting catchment characteristics but then these were not further explained.

Performance criteria – The choice of performance criteria needs to be better justified as this has a large impact on the sensitivity of your results.

Minor Comments

Abstract P1 L6 'Two different flavors of HBV' – this doesn't make sense to me. It would be better to just say two different formulations or types.

P3 L20 'illustrates the frame of these four datasets' – again, this sentence doesn't make sense to me and needs rewriting.

Figure 6 As you are focusing on higher flows, I would also find it useful to have another plot (or combined with Figure 6) that focuses on the flow duration curve for flows higher than the 10th percentile of flow.

Figures 7 -10 need some improvement. The colour scheme needs to be changed in these plots so it is easier for the reader to distinguish between the different catchments.

Currently it is difficult to pick out differences between catchments.

---

## Referee Comment (RC2) · Anonymous Referee #2 · 17 Dec 2018

- Summary

Overall, this is a very interesting paper that approaches the issues of the combined impact of temporal and spatial resolutions on the efficiency of a hydrological model, by using both distributed and lumped versions of the same hydrological model, and different densities and time resolutions of precipitation. I find it however unnecessarily complex, the authors should not try to show us everything they have done, they should try to simplify it into a coherent ensemble. I suggest removing the part on the different rainfall densities, and only keeping the densest network (high density daily disaggregated into hourly). This will allow the authors to focus on the spatial and temporal resolution issues. Also, I suggest to widen the scope of the analysis, which only focuses on high flows presently (because of the chosen criterion).

- Literature issues

I would say that you literature review is quite superficial. Of course, given the considerable increase of published literature, it has become obviously impossible to read everything that is published on a given topic. However, when you aim to publish a paper in a given journal... you should perhaps try to look at what has been published there in more detail. It is a little annoying that you seem to ignore a paper that is precisely on the topic you address in your paper:

Lobligeois, F., V. Andréassian, C. Perrin, P. Tabary, & C. Loumagne. 2014. When does higher spatial resolution rainfall information improve streamflow simulation? An evaluation on 3620 flood events. Hydrology and Earth System Sciences, 18: 575-594

And this is a pity because when you write that "the increase of spatial resolution improved the performance of the model insubstantially or only marginally for most of the study catchments", this is precisely what Lobligeois et al. find...

- Vocabulary issues

I understand that you use "pluviometer" for "recording pluviometer / raingage" and "daily station" for "non-recording pluviometer / raingage". This makes your paper difficult to follow.

- Redaction issues

Your conclusion (especially the last paragraph) is difficult to understand. Try to be more explicit.

- Performance criteria

By using the Nash and Sutcliffe criterion on non-transformed flows (instead of, for example the NS on the square-root or the log or the inverse of flows) you make an explicit

choice to focus on high flows only. Why? Could you extend your study by using another transformation in addition?

- Interception

I would like to know how the interception process is accounted for in your version of HBV? This is important for your comparison, because the simple solutions that work well at the daily time step (i.e. neutralisation of daily rainfall by daily pot. Evaporation) may not work as well at the hourly time step, which may require an interception store.

- Typos

There are a few typos in the paper. Please make a careful check.

---

## Author Comment (AC1) · 14 Jan 2019

We sincerely appreciate Referee 1 for the review of the paper "Sensitivity of hydrological model to the temporal and spatial resolutions of rainfall input". We have considered the reviewer's comments and will revise our manuscript according to the suggestions. The detailed answers to the comments are presented as below.
* * *
Major Comments
* * *
**Introduction – The introduction is quite short and I don't think gives the reader a thorough overview of previous literature on this topic and where this research sits within the field. There have been lots of other studies that have focused on the impacts of spatial and temporal resolution of rainfall on hydrological model output and you need to clearly explain how your research builds on these previous studies. I found it difficult to identify from the introduction what the research gap was and how this study addressed that research gap.**

Response: Thank you for the comments. We will rewrite the literature review for the manuscript. The revised version will contain an updated introduction, referring to the ongoing progress of the study for the sensitivity analysis of rainfall data to model performance both on temporal and spatial scales. We will describe in more details about the attempts for improving model performance and the monition of our study. We will also compare and discuss our idea with previous work on impacts of input variables in hydrological models.

**Study area and hydrometeorological datasets – The rationale for your choice of catchments needs to be outlined. Why were these four catchments chosen? Do they have different climatological characteristics that make them interestingly different? A lot of the following analysis focuses on differences between these mesoscale catchments so it is important that the reader understands what these key differences are. Table 1 contained some interesting catchment characteristics but then these were not further explained.**

Response: Thank you for the constructive comments. According to available flow records, four upstream catchments which are minimally impacted by human influences were considered in this study. These four catchments ranging in size from 417 km$^2$ to about 1300 km$^2$, along with a large difference in elevation and annual precipitation. Meanwhile, the map for raingauge locations also shows different observation density for them. We will further describe the study catchments in the revised paper.

**Performance criteria – The choice of performance criteria needs to be better justified as this has a large impact on the sensitivity of your results.**

Response: We agree that model performance depends strongly on the performance criteria used in calibration. In our previous study, we compared the lumped HBV model performance for difference objective functions in a number of catchments on daily scale. Three criteria: (1) the Nash-Sutcliffe (*NS*), (2) Kling-Gupta efficiency (*GK*) that accounts for the water balances and the correlation of observed and simulated discharge series(Gupta et al., 2009), (3) the combination of *NS* and the *NS* of logarithm of the discharge (*NS+LNS*), were used to calibrate the HBV for 15 catchments(Bárdossy et al., 2016). The model parameters calibrated for every catchment were used to simulate the remaining 14 catchments for testing the transferability of parameters. As shown in the figure below, results for different performance criteria differ considerably. The difference of model performance for the performance measures can be explained by different focuses: *NS* is mainly focusing on high flows as it represents the squared difference between the observed and discharge series, *GK* focuses on water balances and good timing, and *NS+LNS* criterion is strongly influenced by low flow events. Model behavior is dependent on how one evaluates the performance of the model. From the matrix we could find that the model performance for different criteria shows similar treads. In this study, we hope to investigate the sensitivity of model to the input variables and sequentially find effective way for increasing accuracy of flood prediction. We pay for attention to high flows, therefore *NS* was selected as objective function to evaluate model performance. We will add the discussion of the choice of performance measures in conclusion.

[Figure]

Figure. Color-coded matrices for the model performance of parameter transfer for 15 catchments using three difference performance criteria.
* * *
Minor Comments
* * *
**Abstract P1 L6 'Two different flavors of HBV' – this doesn't make sense to me. It would be better to just say two different formulations or types.**

Response: Thanks for the suggestion. We replaced "Two different flavors of HBV" with "two different types of HBV".

**P3 L20 'illustrates the frame of these four datasets' – again, this sentence doesn't make sense to me and needs rewriting.**

Response: Revision made. We replaced this sentence with "Figure 3 shows the flow chart of the data collection and process".

**Figure 6 As you are focusing on higher flows, I would also find it useful to have another plot (or combined with Figure 6) that focuses on the flow duration curve for flows higher than the 10th percentile of flow.**

Response: We appreciate the referee's suggestion, and will add the flow duration curve for flows higher than the 10th percentile of flow (Figure 6(b)) in the revised manuscript.

[Figure]

Figure 6(b). Comparison of the flow duration curve for flows higher than the 10th percentile of flow.

**Figures 7 -10 need some improvement. The colour scheme needs to be changed in these plots so it is easier for the reader to distinguish between the different catchments. Currently it is difficult to pick out differences between catchments.**

Response: Thanks for the suggestion. We have changed the colour of the plots for Figure 7-10 and Figure 12 to make a clear distinction between catchments.

[Figure]

Figure 7. Comparison of NS model performance for using hourly and daily variables as model input for the SH and SD sets.

[Figure]

Figure 8. Comparison of NS model performance for using hourly and daily variables as model input for the DH and DD sets.

[Figure]

Figure 9. Comparison of model performance for different density of rainfall observation network, models were simulated based on daily time step.

[Figure]

Figure 10. Comparison of model performance for different density of rainfall observation network, models were simulated based on hourly time step.

[Figure]

Figure 12. Comparison of model performance for different spatial resolution of model structure.

References:

Bárdossy, A., Huang, Y., and Wagener, T.: Simultaneous calibration of hydrological models in geographical space, Hydrology & Earth System Sciences Discussions, 12, 11223-11268, 2016.

Gupta, H. V., Kling, H., Yilmaz, K. K., and Martinez, G. F.: Decomposition of the mean squared error and NSE performance criteria: Implications for improving hydrological modelling, Journal of Hydrology, 377, 80-91, 2009.

---

## Author Comment (AC2) · 14 Jan 2019

First of all, we would like to sincerely thank Referee 2 for his thorough review of the manuscript and constructive suggestions. The responses to the questions are presented as below.

**- Summary**

**Overall, this is a very interesting paper that approaches the issues of the combined impact of temporal and spatial resolutions on the efficiency of a hydrological model, by using both distributed and lumped versions of the same hydrological model, and different densities and time resolutions of precipitation. I find it however unnecessarily complex, the authors should not try to show us everything they have done, they should try to simplify it into a coherent ensemble. I suggest removing the part on the different rainfall densities, and only keeping the densest network (high density daily disaggregated into hourly). This will allow the authors to focus on the spatial and temporal resolution issues. Also, I suggest to widen the scope of the analysis, which only focuses on high flows presently (because of the chosen criterion).**

Response: There are two main reasons that we presented the results based on different spatial resolutions in the manuscript. Firstly, results indicate the insensitivity of model performance to different spatial resolutions of rainfall for the study catchments, the increase of spatial resolution improved the simulation insubstantially. Secondly, compared to the idea of increasing spatial resolution of model inputs, which causes the complexity of model structure and parameters, using higher temporal resolution of rainfall by disaggregation method could be an easier and much lower cost way to improve model performance. The authors hope to keep the results based on different spatial resolutions of rainfall to emphasize the effects of disaggregation method in model improvement. In the revised manuscript, we will extend the discussion with the sensitivity analysis of model simulation to the choice of performance criteria.

**- Literature issues**

**I would say that your literature review is quite superficial. Of course, given the considerable increase of published literature, it has become obviously impossible to read everything that is published on a given topic. However, when you aim to publish a paper in a given journal: : : you should perhaps try to look at what has been published there in more detail. It is a little annoying that you seem to ignore a paper that is precisely on the topic you address in your paper:**

**Lobligeois, F., V. Andréassian, C. Perrin, P. Tabary, & C. Loumagne. 2014. When does higher spatial resolution rainfall information improve streamflow simulation? An evaluation on 3620 flood events. Hydrology and Earth System Sciences, 18: 575-594 And this is a pity because when you write that "the increase of spatial resolution improved the performance of the model insubstantially or only marginally for most of the study catchments", this is precisely what Lobligeois et al. find…**

Response: We thank the referee for the comments and apology for the ignorance of the references. We will rewrite the literature review part with an updated introduction, referring to the ongoing progress of the researches for the sensitivity analysis of model inputs both on temporal and spatial scales. In the revised version, we will describe in more details about the attempts for improving model performance and the motivation of our paper. We will also compare and discuss our idea with previous work on impacts of input variables in hydrological models.

**- Vocabulary issues**

**I understand that you use "pluviometer" for "recording pluviometer / raingage" and "daily station" for "non-recording pluviometer / raingage". This makes your paper difficult to follow.**

Response: We will replace "pluviometer" with "sub-daily station" in the revised manuscript.

**- Redaction issues**

**Your conclusion (especially the last paragraph) is difficult to understand. Try to be more explicit.**

Response: We will reorganize our conclusion part in the revised manuscript to make it more understandable.

**- Performance criteria**

**By using the Nash and Sutcliffe criterion on non-transformed flows (instead of, for example the NS on the square-root or the log or the inverse of flows) you make an explicit choice to focus on high flows only. Why? Could you extend your study by using another transformation in addition?**

Response: The aim of this work is to investigate the sensitivity of model to rainfall data and sequentially find effective way for increasing accuracy of flood prediction. We pay for attention to high flows so the Nash-Sutcliffe efficiency was selected as objective function to evaluate model performance. In our previous study, we have compared the lumped HBV model performance for difference objective functions in a number of catchments on daily scale. Three criteria: (1) the Nash-Sutcliffe (*NS*), (2) Kling-Gupta efficiency (*GK*) that accounts for the water balances and the correlation of observed and simulated discharge series(Gupta et al., 2009), (3) the combination of NS and the NS of logarithm of the discharge (*NS+LNS*), were used to evaluate HBV for 15 catchments(Bárdossy et al., 2016). In addition, the model parameters calibrated for every catchment were used to simulate the remaining 14 catchments for testing the transferability of parameters. As shown in the figure below, results for different performance criteria differ considerably. The difference of model performance for the performance measures can be explained by different focuses: *NS* is mainly focusing on high flows as it represents the squared difference between the observed and discharge series, *GK* focuses on water balances and good timing, and N*S+LNS* criterion is strongly influenced by low flow events. Model behavior is dependent on how one evaluates the performance of the model. From the matrix we could also find that the model performance for different criteria shows similar treads. In this study, each calibration

process requires 90000 running of HBV model to obtain 10000 best parameter sets. Due to the heavy computation, it is a little bit difficult to extend the study by using some other performance criteria within a short time. We will add the discussion of the choice of performance measures in the conclusion part.

[Figure]

Figure. Color-coded matrices for the model performance of parameter transfer for 15 catchments using three difference performance criteria.

**- Interception**

**I would like to know how the interception process is accounted for in your version of HBV? This is important for your comparison, because the simple solutions that work well at the daily time step (i.e. neutralization of daily rainfall by daily pot. Evaporation) may not work as well at the hourly time step, which may require an interception store.**

Response: In our model, the interception process is consisted in evapotranspiration. The approach of Penman equation (Penman, 1948) is used to estimate the daily potential evapotranspiration according to the long-term monthly mean air temperature and long-term monthly average potential evapotranspiration using observed daily average temperature. Due to the limitation of observed hourly temperature, air temperature and potential evapotranspiration were assumed to be constant over the whole day in our study. The actual evapotranspiration is calculated based on the available water in soil and permanent wilting point based on the

**- Typos**

**There are a few typos in the paper. Please make a careful check.**

Response: We sincerely apologize and will carefully review our manuscript.

References:

Bárdossy, A., Huang, Y., and Wagener, T.: Simultaneous calibration of hydrological models in geographical space, Hydrology & Earth System Sciences Discussions, 12, 11223-11268, 2016.

Gupta, H. V., Kling, H., Yilmaz, K. K., and Martinez, G. F.: Decomposition of the mean squared error and NSE performance criteria: Implications for improving hydrological modelling, Journal of Hydrology, 377, 80-91, 2009.

Penman, H. L.: Natural evaporation from open water, bare soil and grass, Proceedings of the Royal Society of London. Series A.Mathematical and Physical Sciences, 193, 120–145, 1948.

---

## Editor Comment (EC1) · Schaefli (Editor) · 28 Jan 2019

Both reviewers conclude that this is a potentially interesting paper but that it requires major modifications before publication in HESS. This namely includes better motivation of some methodological choices but also an improved literature review, an improve presentation of the results and a conciser formulation of several parts (e.g. the conclusion section). I invite the authors to revise their paper along the lines of their public discussion.

---

## Author Response (AR2)

**Point-by-point response to the Referees**

We would like to sincerely thank the anonymous reviewers and editor for their review of the manuscript. We have considered all the comments and provided detailed responses of how each comment has been addressed in the revised manuscript below (authors' changes in the manuscript are shown in red).
* * *
**1. Response to Anonymous Referee #1**
* * *
Major Comments:

**Introduction – The introduction is quite short and I don't think gives the reader a thorough overview of previous literature on this topic and where this research sits within the field. There have been lots of other studies that have focused on the impacts of spatial and temporal resolution of rainfall on hydrological model output and you need to clearly explain how your research builds on these previous studies. I found it difficult to identify from the introduction what the research gap was and how this study addressed that research gap.**

Response: Thank you for the comments. We have rewritten the literature review for the manuscript. The revised version contains an updated introduction, referring to the ongoing progress of the study for the sensitivity analysis of rainfall data to model performance both on temporal and spatial scales. We described in more details about the attempts for improving model performance and the monition of our study. We compared and discussed our idea with previous work on impacts of input variables in hydrological models.

[revised manuscript text omitted]

**Study area and hydrometeorological datasets – The rationale for your choice of catchments needs to be outlined. Why were these four catchments chosen? Do they have different climatological characteristics that make them interestingly different? A lot of the following analysis focuses on differences between these mesoscale catchments so it is important that the reader understands what these key differences are. Table 1 contained some interesting catchment characteristics but then these were not further explained.**

Response: Thank you for the constructive comments. According to available flow records, four upstream catchments which are minimally impacted by human influences were considered in this study. These four catchments ranging in size from 417 $km^2$ to about 1300 $km^2$, along with a large difference in elevation and annual precipitation. Meanwhile, the map for rain gauge locations also shows different observation density for them. We added more describe the study catchments in the revised paper.

These catchments ranging in size from 417$km^2$ to about 1300$km^2$, along with a large difference in elevation and annual precipitation.   It can be seen clearly from the map that these four catchments have different rain gauge density, the Schwaibach catchment, which located in the mountain area with various elevations (from 190m to 1028m), has the lowest density of rain gauge network and the highest annual precipitation. Rottweil and Kocherstetten have similar climate conditions in terms of annual precipitation and runoff, but the catchment size of Kocherstetten is almost three times of Rottweil. Pforzheim has the smallest drainage area and the lowest amount of precipitation.

**Performance criteria – The choice of performance criteria needs to be better justified as this has a large impact on the sensitivity of your results.**

Response: We agree that model performance depends strongly on the performance criteria used in calibration. In our previous study, we compared the lumped HBV model performance for difference objective functions in a number of catchments on daily

scale. Three criteria: (1) the Nash-Sutcliffe (*NS*), (2) Kling-Gupta efficiency (*GK*) that accounts for the water balances and the correlation of observed and simulated discharge series(Gupta et al., 2009), (3) the combination of *NS* and the *NS* of logarithm of the discharge (*NS+LNS*), were used to calibrate the HBV for 15 catchments(Bárdossy et al., 2016). The model parameters calibrated for every catchment were used to simulate the remaining 14 catchments for testing the transferability of parameters. As shown in the figure below, results for different performance criteria differ considerably. The difference of model performance for the performance measures can be explained by different focuses: *NS* is mainly focusing on high flows as it represents the squared difference between the observed and discharge series, *GK* focuses on water balances and good timing, and *NS+LNS* criterion is strongly influenced by low flow events. Model behavior is dependent on how one evaluates the performance of the model. From the matrix we could find that the model performance for different criteria shows similar treads. In this study, we hope to investigate the sensitivity of model to the input variables and sequentially find effective way for increasing accuracy of flood prediction. We pay for attention to high flows, therefore *NS* was selected as objective function to evaluate model performance. We added the discussion of the choice of performance measures in the manuscript.

[Figure]

Figure. Color-coded matrices for the model performance of parameter transfer for 15 catchments using three difference performance criteria.

Previous studies have shown that model performance strongly depends on the selection of performance criteria (Gupta et al., 2009). The simulated result and model parameters using different objective functions differ considerably as they have different focus (Bardossy et al., 2016). The purpose of this study is to investigate the sensitivity of conceptual model to rainfall variability, and according find effective ways to improve the precision of flood forecasting. Since high flow is extremely important for floods, the Nash-Sutcliffe (NS) efficiency coefficient (Nash and Sutcliffe, 1970) was

used in this study to assess the model performance based on observed discharge. NS efficiency is one of the most widely used performance criteria in model simulation. It focuses on high flow as it evaluates the squared difference between simulated and measured streamflow. NS efficiency can be calculated using the following equation:

This study focuses on high flows and uses only the Nash-Sutcliffe efficiency as the objective function to investigate the model sensitivity. As model performance highly depends on the selection of objective functions, the model sensitivity can be different if using different performance criteria. In addition, all the hourly simulated runoff was aggregated into daily, the hydrological response was evaluated based on daily discharge. Sub-daily response of a catchment is more sensitive to the temporal and spatial variability of rainfall, which could be considered in the future if the hourly discharge observation is available.

Minor Comments:

**Abstract P1 L6 'Two different flavors of HBV' – this doesn't make sense to me. It would be better to just say two different formulations or types.**

Response: Thanks for the suggestion. We replaced "Two different flavors of HBV" with "two different types of HBV".

**P3 L20 'illustrates the frame of these four datasets' – again, this sentence doesn't make sense to me and needs rewriting.**

Response: Revision made. We replaced this sentence with "Figure 3 shows the flow chart of the data collection and process".

**Figure 6 As you are focusing on higher flows, I would also find it useful to have another plot (or combined with Figure 6) that focuses on the flow duration curve for flows higher than the 10th percentile of flow.**

Response: We appreciate the referee's suggestion, and have added the flow duration curve for flows higher than the 10th percentile of flow (Figure 6(b)) in the revised manuscript.

[Figure]

Figure 6(b). Comparison of the flow duration curve for flows higher than the 10th percentile of flow.

**Figures 7 -10 need some improvement. The colour scheme needs to be changed in these plots so it is easier for the reader to distinguish between the different catchments. Currently it is difficult to pick out differences between catchments.**

Response: Thanks for the suggestion. We have changed the colour of the plots for Figure 7-10 and Figure 12 to make a clear distinction between catchments.

[Figure]

Figure 7. Comparison of NS model performance for using hourly and daily variables as model

input for the SH and SD sets.

[Figure]

Figure 8. Comparison of NS model performance for using hourly and daily variables as model input for the DH and DD sets.

[Figure]

Figure 9. Comparison of model performance for different density of rainfall observation network, models were simulated based on daily time step.

[Figure]

Figure 10. Comparison of model performance for different density of rainfall observation network, models were simulated based on hourly time step.

[Figure]

Figure 12. Comparison of model performance for different spatial resolution of model structure.
* * *
**2. Response to Anonymous Referee #2.**
* * *
**- Summary**

**Overall, this is a very interesting paper that approaches the issues of the combined**

**impact of temporal and spatial resolutions on the efficiency of a hydrological model,**

**by using both distributed and lumped versions of the same hydrological model, and different densities and time resolutions of precipitation. I find it however unnecessarily complex, the authors should not try to show us everything they have done, they should try to simplify it into a coherent ensemble. I suggest removing the part on the different rainfall densities, and only keeping the densest network (high density daily disaggregated into hourly). This will allow the authors to focus on the spatial and temporal resolution issues. Also, I suggest to widen the scope of the analysis, which only focuses on high flows presently (because of the chosen criterion).**

Response: There are two main reasons that we presented the results based on different spatial resolutions in the manuscript. Firstly, results indicate the insensitivity of model performance to different spatial resolutions of rainfall for the study catchments, the increase of spatial resolution improved the simulation insubstantially. Secondly, compared to the idea of increasing spatial resolution of model inputs, which causes the complexity of model structure and parameters, using higher temporal resolution of rainfall by disaggregation method could be an easier and much lower cost way to improve model performance. The authors hope to keep the results based on different spatial resolutions of rainfall to emphasize the effects of disaggregation method in model improvement. In the revised manuscript, we have extended the discussion with the sensitivity analysis of model simulation to the choice of performance criteria.

This study focuses on high flows and uses only the Nash-Sutcliffe efficiency as the objective function to investigate the model sensitivity. As model performance highly depends on the selection of objective functions, the model sensitivity can be different if using different performance criteria. In addition, all the hourly simulated runoff was aggregated into daily, the hydrological response was evaluated based on daily discharge. Sub-daily response of a catchment is more sensitive to the temporal and spatial variability of rainfall, which could be considered in the future if the hourly discharge observation is available.

**- Literature issues**

**I would say that your literature review is quite superficial. Of course, given the considerable increase of published literature, it has become obviously impossible to read everything that is published on a given topic. However, when you aim to publish a paper in a given journal: : : you should perhaps try to look at what has been published there in more detail. It is a little annoying that you seem to ignore a paper that is precisely on the topic you address in your paper:**

**Lobligeois, F., V. Andréassian, C. Perrin, P. Tabary, & C. Loumagne. 2014. When does higher spatial resolution rainfall information improve streamflow simulation? An evaluation on 3620 flood events. Hydrology and Earth System Sciences, 18: 575-594 And this is a pity because when you write that "the increase of spatial resolution improved the performance of the model insubstantially or only marginally for most of the study catchments", this is precisely what Lobligeois et al. find…**

Response: We thank the referee for the comments and apology for the ignorance of the references. We rewrote the literature review part with an updated introduction, referring to the ongoing progress of the researches for the sensitivity analysis of model inputs both on temporal and spatial scales. In the revised version, we described in more details about the attempts for improving model performance and the motivation of our paper. We also compared and discussed our idea with previous work on impacts of input variables in hydrological models.

[revised manuscript text omitted]

**- Vocabulary issues**

**I understand that you use "pluviometer" for "recording pluviometer / raingage" and "daily station" for "non-recording pluviometer / raingage". This makes your paper difficult to follow.**

Response: We replaced "pluviometer" with "sub-daily station" in the revised manuscript.

**- Redaction issues**

**Your conclusion (especially the last paragraph) is difficult to understand. Try to be more explicit.**

Response: We reorganized our conclusion part in the revised manuscript to make it more understandable.

In this study, we investigated the impacts of temporal and spatial variability of rainfall in model simulation and parameter estimation. We also explored the question whether higher temporal and spatial resolutions of rainfall lead to any improvement of model performance. Both the lumped HBV and spatially distributed HBV models were applied to simulate the daily runoff for four mesoscale catchments driven by four different types of precipitation datasets which were constructed using a combination of data from high density of daily stations and relatively low density sub-daily stations. The impacts of rainfall variability on model simulation were evaluated using Nash-Sutcliffe efficiency and the mean squared error of flows higher than the 10th percentile of flow. The sensitivity of model to the temporal and spatial resolutions of rainfall was compared. In additional, the common calibration approach was proposed to calibrate the models with different time steps simultaneously for seeking robust model parameters.

For the study catchments, the results indicate that the temporal variability of rainfall data has direct impact on dynamic response of a catchment. For both lumped

and spatially distributed models, if the observation density is the same, the hourly based simulation completely outperforms the daily based simulation, indicating that higher temporal resolution could significantly improve the model performance. Disaggregating high density daily observations into relatively low density sub-daily values could lead to considerable model improvement, especially for the catchment with a sparse rain gauge network. Rainfall disaggregating approach provides an effective way for increasing the temporal resolution of rainfall and the performance of model simulation. However, the lumped and spatially distributed HBV model perform very similarly, indicating that higher model resolution does not or only marginally improve the model performance for the study catchments. The result supports the general findings of Lobligeois et al. (2014) and Zhu et al. (2018), where insignificant improvement was observed using higher spatial resolution of rainfall. The reason that the spatially distributed model does not outperform the lumped model could be due to the fact the study catchments are smaller than 2000km2 and have relatively uniform precipitation.

As discussed at the beginning of this paper, we aim to investigate the sensitivity of model to rainfall variability and to find effective ways for improving the model performance. This research indicates that data disaggregation approach could lead to a significant improvement of model performance, while higher spatial resolution of rainfall does not always enhance model performance. Most of the hydrological models can be easily adjusted to use different time steps. The study suggests that increasing the temporal resolution of precipitation inputs with disaggregation method could be an easier and more efficient to improve model performance, compared with increasing the model spatial resolution that comes at a cost of increasing the complexity of model structure and parameters.

This study focuses on high flows and uses only the Nash-Sutcliffe efficiency as the objective function to investigate the model sensitivity. As model performance highly depends on the selection of objective functions, the model sensitivity can be different if using different performance criteria. In addition, all the hourly simulated runoff was aggregated into daily, the hydrological response was evaluated based on daily

**- Performance criteria**

**By using the Nash and Sutcliffe criterion on non-transformed flows (instead of, for example the NS on the square-root or the log or the inverse of flows) you make an explicit choice to focus on high flows only. Why? Could you extend your study by using another transformation in addition?**

Response: The aim of this work is to investigate the sensitivity of model to rainfall data and sequentially find effective way for increasing accuracy of flood prediction. We pay for attention to high flows so the Nash-Sutcliffe efficiency was selected as objective function to evaluate model performance. In our previous study, we have compared the lumped HBV model performance for difference objective functions in a number of catchments on daily scale. Three criteria: (1) the Nash-Sutcliffe (*NS*), (2) Kling-Gupta efficiency (*GK*) that accounts for the water balances and the correlation of observed and simulated discharge series(Gupta et al., 2009), (3) the combination of NS and the NS of logarithm of the discharge (*NS+LNS*), were used to evaluate HBV for 15 catchments(Bárdossy et al., 2016). In addition, the model parameters calibrated for every catchment were used to simulate the remaining 14 catchments for testing the transferability of parameters. As shown in the figure below, results for different performance criteria differ considerably. The difference of model performance for the performance measures can be explained by different focuses: *NS* is mainly focusing on high flows as it represents the squared difference between the observed and discharge series, *GK* focuses on water balances and good timing, and N*S+LNS* criterion is strongly influenced by low flow events. Model behavior is dependent on how one evaluates the performance of the model. From the matrix we could also find that the model performance for different criteria shows similar treads. In this study, each calibration process requires 90000 running of HBV model to obtain 10000 best parameter sets.

Due to the heavy computation, it is a little bit difficult to extend the study by using some other performance criteria within a short time. We added the discussion of the choice of performance measures in the revised paper.

[Figure]

Figure. Color-coded matrices for the model performance of parameter transfer for 15 catchments using three difference performance criteria.

Previous studies have shown that model performance strongly depends on the selection of performance criteria (Gupta et al., 2009). The simulated result and model parameters using different objective functions differ considerably as they have different focus (Bardossy et al., 2016). The purpose of this study is to investigate the sensitivity of conceptual model to rainfall variability, and according find effective ways to improve the precision of flood forecasting. Since high flow is extremely important for floods, the Nash-Sutcliffe (NS) efficiency coefficient (Nash and Sutcliffe, 1970) was used in this study to assess the model performance based on observed discharge. NS efficiency is one of the most widely used performance criteria in model simulation. It focuses on high flow as it evaluates the squared difference between simulated and measured streamflow. NS efficiency can be calculated using the following equation:

This study focuses on high flows and uses only the Nash-Sutcliffe efficiency as the objective function to investigate the model sensitivity. As model performance highly depends on the selection of objective functions, the model sensitivity can be different if using different performance criteria.

**- Interception**

**I would like to know how the interception process is accounted for in your version of HBV? This is important for your comparison, because the simple solutions that work well at the daily time step (i.e. neutralization of daily rainfall by daily pot.**

**Evaporation) may not work as well at the hourly time step, which may require an interception store.**

Response: In our model, the interception process is consisted in evapotranspiration. The approach of Penman equation (Penman, 1948) is used to estimate the daily potential evapotranspiration according to the long-term monthly mean air temperature and long-term monthly average potential evapotranspiration using observed daily average temperature. Due to the limitation of observed hourly temperature, air temperature and potential evapotranspiration were assumed to be constant over the whole day in our study. The actual evapotranspiration is calculated based on the available water in soil and permanent wilting point based on the

**- Typos**

**There are a few typos in the paper. Please make a careful check.**

Response: We sincerely apologize and have carefully reviewed our manuscript.

**List of additional references:**

[revised manuscript text omitted]

---

## Author Response (AR3)

**Point-by-point response to Reviewers**

Dear editor and reviewers,

We would like to thank all of you for the review of our manuscript and the constructive suggestions. All the comments have been considered and a point by point response has been provided below.

For Referee 2's additional question about the structure of the manuscript, we provided a further explanation on why we prefer to keep the subsection describing the impact of spatial resolution of rainfall on model performance. The manuscript has been thoroughly revised and polished carefully with the reviewers' help.

The point-by point response is formatted as follows:

- the referees' comments are shown in black

- authors' response are shown in blue
* * *
1. Response to Anonymous Referee #1
* * *
The authors have satisfactorily addressed my reviewer comments and have made a number of revisions which have improved the manuscript. I recommend that the manuscript is accepted with a few minor revisions detailed below.

1. The introduction is much improved with a wider range of references but the English needs to be carefully checked as there are a couple of sentences that don't make sense as currently written. Specifically:

a. P3 L1-3 "As most of the hydrological models are flexible and can be easily adjusted to different time steps, which makes the sensitivity analysis of model output to the temporal variability of rainfall easy."

b. P3 L13-15 "This could properly lead to a better understanding of the sensitive of rainfall inputs and help to identify relatively economical ways to improve tremendously the model behavior."

Response: We sincerely thank you for the valuable comments. We have rewritten the above sentences to make them easier to be understood. The revised manuscript has been sent to two professionals for proofreading. According to their suggestions, we have thoroughly corrected the grammar and improved the clarity of the sentences. All the corrections are marked in the revised

version.

2. I liked the addition of Figure 7 but I think the x-axis is wrong. If you have plotted simply the top 10th percentile of flows then surely the x-axis should go from 0 to 10 rather than 0 to 100? Response: Thank you for point it out. Yes, the x-axis should go from 0 and 10. We have added the corrected flow duration curve for flows higher than the 10th percentile of flow in the revised manuscript.

Figure 7. Comparison of the flow duration curve for flows higher than the 10th percentile of flow.

3. Figures 8 - 11 - although the authors have made changes to these plots, I still find it difficult to distinguish between the colours (particularly the red and pink dots). It would be worth changing the pink dot to a green dot so it is easier to identify the catchments.

Response: Thanks for the suggestion. We have changed the data point color schemes in Figure 8-11 (change the pink and cyan dots to green and yellow, respectively). As shown in Figure 8, it can be easier to identify the catchments than the original figure.

Figure 8. Comparison of NS model performance for using hourly and daily rainfall as model input for the DH and DD sets.

**2. Response to Anonymous Referee #2**

The literature review was extended, that is a good point. However it now reads like a very long list and the authors should help the readers by adding a few summarizing sentences, to underline that the literature does not have a "ready" answer for the question they ask and even that the papers they review do not agree with each other (which is a further reason for writing this paper). Response: We sincerely appreciate you for reviewing our manuscript. Your valuable comments and suggestions led to an improved version of the manuscript. We have reorganized the introduction part, and the literature review is expanded with a summary of previous studies.

As far a the restructuration/simplification of the paper I had suggested the authors did not do it. I understand that this is a lot of work. However, in my opinion, it would have made the paper simpler and easier to understand. I believe the paper still reads more like an exhaustive report than as a selection of the most interesting results.

Response: We thank the reviewer for your suggestion on simplifying the paper structure as mentioned in your previous comments: "I suggest removing the part on the different rainfall densities, and only keeping the densest network (high density daily disaggregated into hourly)". We think that the imperfection of the sub-section titles in the earlier version hinder the reviewer from better understanding the logic flow of our result section. Therefore, we revised the sub-section titles of the result section in the revised manuscript. The revised sub-sections are:

"4.1 Comparison of the rainfall dataset", "4.2 Results of calibration and validation", "4.3 Model performance using different temporal resolutions of rainfall data ", "4.4 Model performance in terms of observation density", "4.5 Model performance in terms of spatial resolution of rainfall data", and "4.6 Common model calibration with different temporal resolutions". As you may tell, the revised sub-section titles are easy to follow and each sub-section are closely related to the objectives and unique to each other. In addition, six sub-sections in a result section are not too many. We therefore believe the current structure works fine.

In addition, we would like to answer in more detail for the questions you raised during your first review of the manuscript:

(1) The purpose of this study is to find the effective ways for improving model performance for flood forecasting. It requires understanding the sensitivity of the rainfall-runoff modes to rainfall input data. The spatial variability of rainfall strongly influences the timing and shape of hydrograph, while the temporal variability mainly affects the peak of flood wave. As increasing the temporal and spatial resolutions of model are two common methods in hydrological modeling, we believe that the comparison of temporal and spatial variability of rainfall to runoff simulation is very important. With the testing of different spatial resolutions, we hope to answer the specific question that which one is more efficient to improve model performance: Increasing temporal resolution or spatial resolution of rainfall? We can conclude from this study that higher temporal resolution of rainfall can lead to a significant improvement of model performance, while higher spatial resolution of rainfall does not always enhance model performance. It suggests that compared with increasing the model spatial resolution that comes at a cost of increasing the complexity of model structure and parameters, increasing the temporal resolution of precipitation inputs with disaggregation method can be easier and more efficient to improve model performance. We think it is worth preserving the results based on different spatial resolutions of rainfall in the manuscript.

(2) This study aims to increasing the accuracy of flood prediction and thus pays more attention on the high flow. The HBV model performance to different performance criteria has been investigated by our previous study. Result shows that the model sensitivity can be different is the model performance is measured differently. Result also shows that for most of the cases, the model performance for different objective functions have same tendency of changes under different catchments. In this study, each calibration process requires 90000 running of HBV model to obtain 10000 best parameter sets. Due to the heavy computation, we only tested the sensitively of model performance based on the most widely used performance criterion- NS coefficient. NS coefficient represents the squared difference between the observed and discharge series and mainly focuses on the high flow, which could satisfy the needs of improving the accuracy of flood prediction.

(3) The HBV model used in this study is relatively simple. There is no interception routine in this version of HBV model. Currently, the process of interception is simulated implicitly in the evapotranspiration part of the model and the comparison of different temporal resolutions is based on the simulation of daily runoff. In our further study, the interception routine will be included to investigate the impact in hourly simulation.

Last, I still found a number of typos (an example the authors write "in additional" instead of "in addition" in the conclusion).

Response: We have carefully proofread the manuscript and corrected the typos and grammar errors in the revised manuscript.

**Sensitivity of hydrological model to thetemporal and spatial resolutions of rainfall inputdata**

Yingchun Huang1, András Bárdossy2, and Ke Zhang1,3

1College of Hydrology and Water Resources, Hohai University, Nanjing 210098, China
2Institute for Modelling Hydraulic and Environmental Engineering, University of Stuttgart, Stuttgart D-70596, Germany
3State Key Laboratory of Hydrology-Water Resources and Hydraulic Engineering, Hohai University, Nanjing 210098, China

**Correspondence:**

Ke Zhang (kzhang@hhu.edu.cn); Yingchun Huang (yingchunhuang@hhu.edu.cn)

Abstract. As Rainfall is the most important input for rainfall-runoff models; precipitation It is usually observed measured at specific sites on a daily or sub-daily time scale and requires interpolation for further application. This study aims to explore that for a given objective function, whether evaluate if a higher temporal and spatial resolution of precipitation could provide an improvement in rainfall can lead to improved model performance. Four different gridded hourly and daily precipitation rainfall datasets ; with a spatial

- 5 resolution of 1km×1km for the state of Baden-Württemberg state ofin Germany were constructed using a combination of data from a dense network of daily rainfall stations and a less dense network of sub-daily stations. Two different types of HBV models with different model structures, A lumped and a spatially distributed HBV models were used to investigate the sensitivity of model performance onto the spatial variability of precipitation. resolution of rainfall. For four selected mesoscale catchments, these The four different rainfall datasets were used to simulate the daily discharges using both lumped and semi-distributed HBV models.drive both lumped and dis-
- 10 tributed HBV models to simulate daily discharges in four catchments. Different possibilities of improving the accuracy of daily streamflow prediction were investigated. Three main results were obtained from this study: Main findings include (1) a higher temporal resolution of precipitation improved rainfall improves the model performance if the observation station density wasis high; (2) a combination of observed high temporal-resolution observations with disaggregated daily precipitation rainfall leads to afurther improvement in the model performance of the tested models; (3) for the present research, the increase of spatial resolution improved improves the
- 15 performance of the model insubstantially or only marginally forin most of the study catchments.

**1 Introduction**

Rainfall is one of the most important driving forces in hydrological modeling and produces a direct impact on a primary driver of hydrological models and can impact catchment runoff response significantly (Obled et al., 1994; Ly et al., 2013). In general, rRainfall is usually measured by standard rain gauges or wireless telemetering pluviometers over a period of time (e.g. daily, sub-daily). The Uncer-

20 tainties in rainfall estimation for a catchment can occur due to instrument measuringerror errors as well as and the representativeness of point rainfall causes a certain amount of uncertainty in precipitation estimation for a specific catchment. The spatial and temporal variability of precipitation rainfall. -is one of The latter are the main sources of uncertainty uncertainties in model simulation and flood forecasting (Beven, 1998; Berne et al., 2004). The spatial variability of rainfall strongly influences the timing and shape of hydrograph, while the temporal variability mainly affects the peak of flood wave (Singh, 1997). Therefore, it is of great significance to investigate the sensitivity of hydrological models to rainfall input and find an effective way to improve the accuracy of model simulation and flood forecasting. The improvement of flood simulation requires understanding the sensitivity of the rainfall-runoff models to rainfall input data. In recentOver the past decades, extensive efforts have been put on investigating the influence of rainfall spatial variability in hydrological models. Different

- 5 interpolationvarious methods have been used to obtain the spatial distribution structuredistributions of rainfall based on rain gauge data and catchment characteristics (Goovaerts, 2000; Jeffrey et al., 2001; Hofierka et al., 2002; Haylock et al., 2008; Ly et al., 2013). These approaches can potentially improve the spatial resolution of rainfall that is used as input for rainfall-runoff models, thereby reducing the uncertainty of hydrological models. Singh (1997) found that the spatial variability of rainfall can have significant influence on the timing and shape of hydrograph, while the temporal variability can affect the peak of flood wave. Kobold and Brilly (2006) used a different number of rain gauge stations to derive areal
- 10 rainfall and quantified uncertainties of rainfall inputs using HBV model in hourly time step. derived hourly areal rainfall interpolated from various numbers of rain gauges to quantitatively assess the sensitivity of peak flow to the uncertainty of rainfall data using an HBV model. They found that the error in precipitation rainfall may lead to even greater error in the peak of flood flood peak. Bardossy and Das (2008) also invested studied the impact of spatial variability of rainfall by varying the distribution of therain gauge network. They found that the transferability transferabilities of model parameters calibrated based on sparesparse and density dense rainfall
- 15 information isare very different. Das et al. (2008) used four different model structures to simulate daily runoff in central Europe. Results indicated that the semi-distributed and semi-lumped models outperform the lumped and distributed model structures, and they naturally concluded. They suggested that the lack of spatial information is responsible for the low efficiency of distributed model. Xu et al. (2013) indicated that the increase of rain gauge network density gradually improves can improve the model performance<del>up to some threshold</del>, but no apparent improvement was observed when the number of rain gauges exceeded
- 20 thea threshold. Lobligeois et al. (2014) investigated the impacts of rainfall spatial variability by implementing diverse representations of model for a considerable number of catchments. They typically found that for the region with variable precipitation, the semi-distributed models outperform the lumped one, but these two models perform similar for the catchments that having relatively uniform precipitation. found that simi-distributed models outperform the lumped models when rainfall is highly variable over simulation catchment, but they perform similarly when rainfall is relatively uniform. Emmanuel et al. (2015) proposed rainfall variability indexes to carefully evaluate characterize the influence of
- 25 rainfallspatial variability rainfall and implemented this approach in the model simulation for the Cevennes catchment in France (Emmanuel et al., 2017). They found that higher spatial resolution of rainfall could achieve better model performance. We can learn from these researches that the sensitivity of model performance to the spatial resolution of rainfall seems different for some of the case studies. However, the increasing of spatial resolution in model simulation leadscan lead to considerable complexity of model structure and requires for require much more data than using a lumped version.
- 30 Simultaneously, the The rainfall-runoff response of a catchment is also strongly impacted by the temporal variability of rainfall (Bárdossy and Pegram, 2016). The high High temporal resolution rainfall data is typically measured by are collected at pluviometer stations (wireless instruments recording at sub-daily intervals, be called sub-daily data in the following), with telemetry at sub-daily time resolutions. which faces the problem of poor dataSub-daily data often have poor quality caused by equipment malfunction or misreading. Compared with sub-daily rainfall data, the daily rainfall datadaily data are more reliable and plentifultend to be more available and
- 35 reliable, cover a longer duration of time periods. Disaggregating daily into sub-daily values data offers a potential solution to

accurately capture the temporal variability of rainfall (Parkes et al., 2013; Bardossy and Pegram, 2016). Pui et al. (2012) properlycompared three different approaches for disaggregating daily rainfall into sub-daily series and indicated found the resampling method is the best wayone for rainfall disaggregating disaggregation. Bárdossy and Pegram (2016) used Gaussian Copula-based model for disaggregating daily data to infill the gap of pluviometersub-daily data, and they found that this conditional disaggreg-

- 5 gation of precipitationrainfall is reliable and applicable in various regions. Breinl and Di Baldassarre (2019) applied a spatial method of fragments to disaggregate daily precipitationrainfall into hourly values. Although considerable studies have been carried out in the interpolation of sub-daily rainfall, thoroughly verification of the data quality of these products through the comparison of rainfall-runoff simulation results is required. It is extreme important to find out if the disaggregation leads to an improvement of model performance. As most of the hydrological models are flexible and can be easily adjusted to different time steps, which makes the sensitivity analysis of model output to the temporal variability of rainfall
- 10 easy.Kobold and Brilly (2006) found that calibrating hydrological models with sub-daily time steps can significantly improve the accuracy of flood forecasting.

Furthermore, certainSome studies focus on both the spatial and temporal resolution of rainfall. Bruneau et al. (1995) indicertedfound that the temporal and spatial resolutions of rainfall used foras the inputs of thehydrological model possess amodels can have considerable influence on the model efficiency and parameters parameter values. Booij (2002) assuredly found that the in-

- 15 fluence of modelrainfall spatial resolution is indeedgreater than rainfalltemporal variability on the resolution in terms of simulation of extreme flowflows. Meselhe et al. (2009) indicated pointed out that the physically based model is models are more sensitive to the spatial and temporal resolution of rainfall data than the conceptual modelmodels. Zhu et al. (2018) found that the spatial variability of rainfall is much more sensitive to model performance for catchments larger than 2000km2 under dry soil condition; while flood, and floods in the small catchments is controlled are more influenced by the temporal variability of rainfall. Since a vast number
- 20 of efforts had been made to improve So far, more efforts have been invested in improving the spatial or temporal resolution of rainfall, it is important to focus on a quantitative analysis and but there are less studies on quantification and direct comparison of the potential effect of rainfall temporal variability with the spatial variability to catchment dynamic response. This could properly lead to a better understanding of the sensitive of rainfall inputs and help to identify relatively economical ways to improve tremendously the model behavior.catchment dynamic responses driven by different rainfall temporal and spatial resolutions.
- The ultimateoverarching aim of this study is to undoubtedly gain more firsthand knowledge on understand the dependency of hydrological model performance on the precipitation rainfall data. The specific research objectives are three-fold: (1) investigate the effects of rainfall data quality on model performance, (2) examine the sensitivity of model performance to different spatial and temporal resolutions of rainfall data using two different model spatial configurations, and (3) explore the possibility of improving model performance on a daily scale. The effects of rainfall data quality on model performance were investigated. The sensitivity of
- 30 model performance to different spatial and temporal resolutions of rainfall data was examined using two distinctive model structures. The possibility of improving model performance on a daily scale was properly discussed. The manuscript is organized as follows: the introduction, paper will be followed by section 2 , which describes to describe the study area and the precipitation rainfall datasets used in this research. In section 3, the hydrological model and the calibration framework used in this research method are explained, while section. Section 4 presents the results and discussion of this work. The conclusions and outlook are provided in section 5.

**2 Study area and hydrometeorological datasets**

30

This study was tested area is located in a semi-humid region in the Baden-Württemberg state of Germany (Figure 1) that characterized by temperate monsoon climate of mild winter and warm summer. Elevations of this state range Elevation of this region ranges from 85m to 1493m above sea level. The heterogeneity of climate characteristics is mainly due to the great

- 5 variability of elevations within the study area. Winters are mild whereas summers are warmer. The annual mean air temperature in Baden-Württemberg is about 10.2 °C. PrecipitationRainfall is evenly distributed throughthroughout the year. However, its seasonality shows a weak trend. The monthly rainfall reaches its peakis highest in June, whereas the month of October shows the least precipitationand lowest in October. The meteorological observationsdata used in this study waswere provided by the German Weather Service (DWD). Daily air temperature data required for the rainfall-runoff model waswere interpolated on a 1×1 km2 grid
- 10 from the observations using the algorithm of External Drift Kriging algorithm (Ahmed and De Marsily, 1987). The topographical elevation was taken as external drift (Hundecha and Bárdossy, 2004; Das et al., 2008). The long term monthly potential evapotranspiration and the average air temperature were used to compute the daily potential evapotranspiration using the Hargreaves and Samani method (Hargreaves and Samani, 1985).
- PrecipitationRainfall data from a dense network of daily precipitationrainfall stations (62 km2/station in 1991) and from a less dense network of sub-daily stations (144 km2/station in 1991) with high resolution precipitationrainfall observations were used for this study. All available data fromdata are available for the time period 1991-2010 was considered. The number of available daily stations and sub-daily stations varies according to different time period. Figure 2 illustrates the number of available observation locationsstations in Baden-Württemberg between the years 1991 and 2010. It can be seen from the graph, more than 430 daily stations were available in 1991, whilebut only 30 sub-daily stations were available in 1991. The total number of daily
- 20 stations decreased dramatically to 250 around 2003 and remained constantstable for the subsequent years. The number of sub-daily stations kepthas been increasing throughout the whole this period and experienced a sharp increase from 100 to 200 in the year 2005. The following Four different precipitation rainfall datasets were created according to the available observed data: generated and explained as follows.
  - 1. High temporal resolution observed precipitation rainfall was aggregated to hourly time steps and then interpolated subse-
- 25  $\frac{1}{1 \times 1}$  km2 gridgrids using the ordinary Kriging algorithm (Matheron, 1963). The correlation function obtained from the cross-correlations of the hourly time series was used as a basis for the variogram. This set will be referred to as Sparse Hourly (SH) set.
  - 2. Observed daily precipitationrainfa